# Programmable de novo designed coiled coil-mediated phase separation in mammalian cells

Maruša Ramšak[1,2], Dominique A. Ramirez[3], Loren E. Hough[4], Michael R. Shirts[5], Sara Vidmar[1,2], Kristina Eleršič Filipič [6], Gregor Anderluh [6] & Roman Jerala [1] ✉

Membraneless liquid compartments based on phase-separating biopolymers have been observed in diverse cell types and attributed to weak multivalent interactions predominantly based on intrinsically disordered domains. The design of liquid-liquid phase separated (LLPS) condensates based on de novo designed tunable modules that interact in a well-understood, controllable manner could improve our understanding of this phenomenon and enable the introduction of new features. Here we report the construction of CC-LLPS in mammalian cells, based on designed coiled-coil (CC) dimer-forming modules, where the stability of CC pairs, their number, linkers, and sequential arrangement govern the transition between diffuse, liquid and immobile condensates and are corroborated by coarse-grained molecular simulations. Through modular design, we achieve multiple coexisting condensates, chemical regulation of LLPS, condensate fusion, formation from either one or two polypeptide components or LLPS regulation by a third polypeptide chain. These findings provide further insights into the principles underlying LLPS formation and a design platform for controlling biological processes.

The process of liquid-liquid phase separation (LLPS) has become recognized as a driver for the formation of membrane-less compartments present in all types of cells[1,2]. Diverse types of biomolecular condensates show similarities in size, shape, dynamics, and manner of assembly, even though they differ in composition, function, and location within the cell[2,3]. They differ from membrane-delimited organelles, especially in their ability to spontaneously form and dissolve, as well as in their permeability[4,5]. The list of cellular compartments thought to involve LLPS is growing, including large structures such as nucleoli, nuclear speckles, P-bodies, as well as smaller assemblies, such as signaling granules, DNA damage foci, and receptor clusters[6–18]. The resulting membraneless structures affect cellular

functions, such as biomolecular sequestration, channeling signaling molecules, and facilitating reactions[6,13,19–29].

LLPS results from a density transition where a solution of macromolecules, such as proteins, separates into a dense phase forming droplets rich in macromolecules, which coexist with a dilute phase[30]. Enrichment with multivalent molecules is a distinct feature of biomolecular condensates; such molecules contain multiple elements that lead to intra- and/or inter-molecular interactions inherently decreasing the solubility of molecules and promoting phase separation[1,2,31]. Adhesive interactions can occur either between intrinsically disordered regions (IDR) providing multiple weakly interacting elements or between oligomerizing folded domains[32–34]. In the case of IDR-

[1]Department of Synthetic Biology and Immunology, National Institute of Chemistry, Ljubljana, Slovenia. [2]Interdisciplinary doctoral study of biomedicine, Medical Faculty, University of Ljubljana, Ljubljana, Slovenia. [3]Department of Biochemistry, University of Colorado Boulder, Boulder, CO, USA. [4]Department of Physics and BioFrontiers Institute, University of Colorado Boulder, Boulder, CO, USA. [5]Department of Chemical and Biological Engineering, University of Colorado Boulder, Boulder, CO, USA. [6]Department of Molecular Biology and Nanobiotechnology, National Institute of Chemistry, Ljubljana, Slovenia. ✉e-mail: roman.jerala@ki.si

containing polypeptides, amino acid motifs that govern the formation of LLPS − such as charge segregation or patterns of aromatic residues −have been proposed, but the structure of interacting motifs is often not known[35–37]. The interaction domains are typically separated by linkers that provide connectivity or structural constraints[38,39]. A stickers and spacers framework has been developed to describe the polypeptide LLPS, where stickers represent the interacting domains and spacers the regions separating them[38,40]. Nevertheless, the minimal components forming LLPS and their interactions are often not well characterized as most designed LLPS have been based on modifications of existing polypeptides or natural protein domains known to phase separate[36,37]. The bottom-up engineering of LLPS based on designed well-understood interacting modules could offer an additional insight into this phenomenon and, moreover, facilitate the introduction of new biological functionality.

Coiled-coil (CC) dimers represent one of the smallest well-understood protein-protein interaction motifs, where two or more peptides interact along a complementary interface combining electrostatic and hydrophobic interactions to form their characteristic helical structure, while in isolation they typically remain unstructured[41,42]. CCs can be designed by implementing of interaction rules that confer specificity, enabling fine-tuning of the interaction strength by varying their sequence and lengths[42]. Coiled coils have been reported in some natural condensates[43–45], and we therefore hypothesize that the tunable multi-valent interactions formed by CCs could be designed to drive LLPS formation, although the rules governing LLPS formation by this mechanism will have to be established.

In this study, we engineered de novo biomolecular condensates in mammalian cells, composed of oligomers of rationally designed CC-dimerizing segments connected by linkers. The design is based on building modules that, through multiple weak interactions between CC-forming segments, lead to the formation of LLPS. In this system, we explored the effect of multivalency, orthogonality of interacting CC segments, interaction strength, sequential order, the orientation of CC dimers, and the impact of polypeptide linkers on condensate formation. We established how the selection of the CC building modules defines the fluidity of condensates. The fundamental hypothesis behind these phase-separating CC proteins is supported by a coarse-grained (CG) molecular dynamics simulation framework with a minimal set of tunable parameters. The simulations robustly predict the formation of condensates driven by CC interactions and are consistent with the experimentally established design principles. We demonstrate the engineering of several features into CC-LLPS, including the chemically inducible formation or dissolution of condensates, the formation of multiple coexisting synthetic condensates in the mammalian cells as well as the design of a single polypeptide chain-based LLPS comprising only two different homodimeric CC pair interactions. This de novo-designed CC-LLPS platform, together with supportive and predictive coarse-grained simulations, offers an insight into the principles of LLPS formation and is a powerful tool for engineering biological systems.

## Results

### Design of a two-chain LLPS condensate based on CC-dimer interactions

We aimed to employ the current consensus understanding of phase separation drivers to design de novo polypeptides that could lead to phase separation and the formation of condensates in mammalian cells based on designable interaction modules. Transient and noncovalent multivalent interactions between molecules are the main drivers for the formation of and defining the properties of condensates[46]. The number and affinity of interacting domains, as well as the linker properties, play a role in LLPS formation[34,39,47]. While the number of interacting modules and the strength of

domain interaction govern phase separation[34,48,49], those parameters must be balanced as high affinity may lead to the formation of immobile aggregates.

We designed an orthologous set of weakly interacting CC pairs, with adjustable affinities suitable for LLPS formation (sequences in Supplementary Table 1). The design was based on the same principles as used previously for the design of strongly interacting orthogonal pairs[50–53]. Canonical parallel CC dimers are composed of heptad repeats, with positions labeled abcdefg. The interaction between pairs of CC helices is driven primarily by a combination of hydrophobic or complementary electrostatic interactions at positions a, d, and e, g, respectively[51]. The design of CC dimers is enabled by the implementation of favorable interactions at these sites, with noninteracting b, c, and f positions permitting fine-tuning of the interaction strength based on their helical propensity and intrachain electrostatic interactions[54,55]. Employing these rules, we designed three sets of 4 peptides, each set containing two orthologous pairs. The orthogonality between CC pairs is based on matching hydrophobic and electrostatic patterns as well as a pattern of Asn/Ile residues at position a[42,51,54,56] that also defines the orientation of the CC dimer (parallel vs. antiparallel). Our initial set of CC segments, labeled 'S', has the weakest affinity with 3-heptad CC modules (S1-S4), where S1 was designed to form a pair with S2 and S3 with S4. At positions b, c we introduced amino acid residues Gln and Ser, to weaken the helical propensity[57], and thus lower the affinity between the CC pairs. We did not observe any helical structure in circular dichroism (CD) spectra, and there was no difference in the spectra of a single peptide versus peptide pairs (Supplementary Fig. 1a, b). In the next set of peptides called Sh, amino acid residues at positions b, c from the previous set were substituted with Ala, inserted to promote the helical propensity of CC (sequences in Supplementary Table 1) and increase the stability. CD spectra showed increased helical content of a peptide pair, compared to a single peptide, especially at low temperatures (Supplementary Fig. 1c, d). The last CC set named Pf, consisting of 4-heptad long CC (P5f-P6f and P13f-P14f), had high helical content in the case of a peptide pair but not for isolated peptides as measured with CD (Supplementary Fig. 1e, f). Furthermore, we showed orthogonality between peptide pairs for Sh and P sets (Supplementary Fig. 1g, h). The binding affinity of peptide pairs was determined using isothermal titration calorimetry (ITC), where 4-heptad modules had affinity in the low micromolar range (P5f-P6f had $K_d$ of $5.9 \pm 0.5\,\mu M$, and P13f-P14f a $K_d$ of $2.3 \pm 0.2\,\mu M$, Supplementary Fig 2). The affinity of three-heptad Sh peptides is expectedly substantially weaker with $K_d$ of $2.3 \pm 1.5\,mM$ for S1h-S2h and $K_d$ of $0.34 \pm 0.07\,mM$ for S3h-S4h (Supplementary Fig. 2). We expect this weak affinity could enable rapid exchange required for the liquid state. The affinity of S peptide pairs was too weak to be measured experimentally by ITC.

To provide multivalent interactions driving phase separation, we designed each polypeptide chain to contain several interacting CC segments acting as stickers[38,40] that are concatenated into a single chain via spacers. Further, to prevent the formation of oligomerization-incompetent straight dimers, a mismatch in the pattern of interacting modules in two polypeptide chains was introduced. All designed polypeptides used in this study, together with their location and schematic are collected in Supplementary Table 2.

If only a single type of the CC-forming peptide was present in each of the two polypeptide chains, no condensates were formed, as they can only form dimer or linear polymers and does not enable formation of a network (Supplementary Fig. 3a, b). We included two different types of CC-forming segments from the above-described designed set in each chain to facilitate interaction between multiple chains. Therefore, polypeptide chains were composed of one to three copies of each of the two three-heptad CC pairs (S1-S2, S3-S4) amounting to a range between two to six CC-forming segments, respectively, with different patterns of CC arrangement in each chain. In the first chain, CC from

**a** Components:

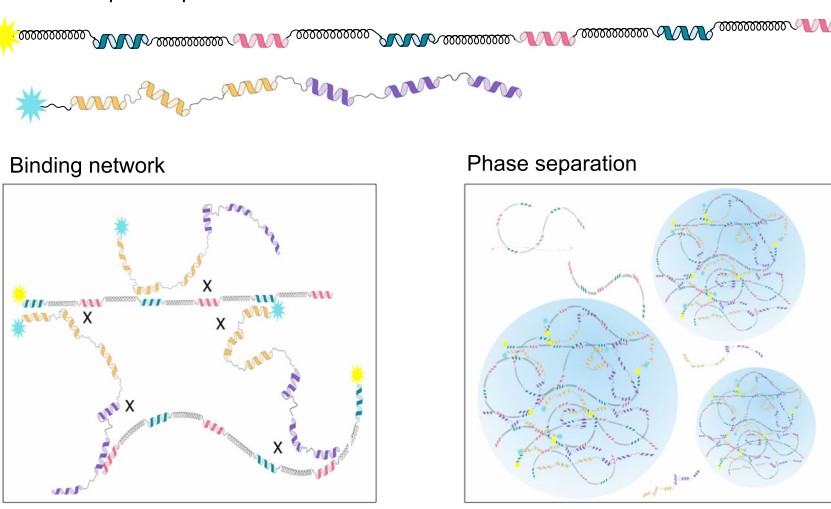

**b** Single protein expressed

**c** Protein pair expressed

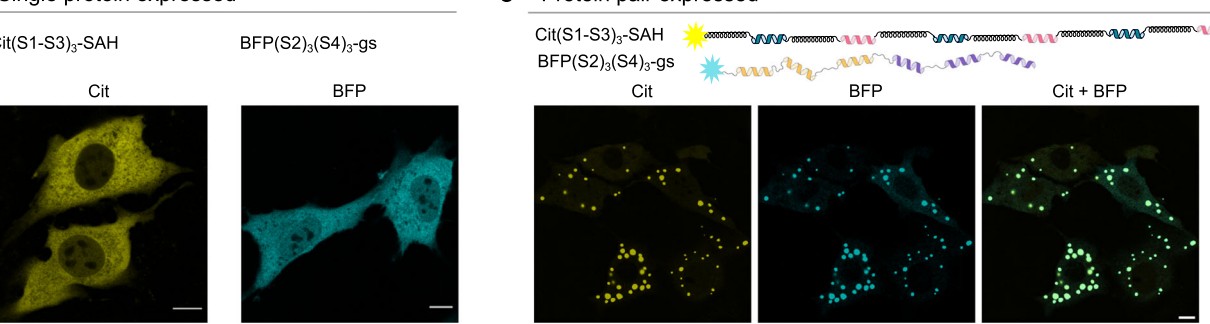

**Fig. 1 | Schematic illustration of CC-dimer-based designed LLPS condensates in mammalian cells. a** Coiled-coil dimer-forming peptides as interaction modules between polypeptide chains. Each polypeptide chain comprises multiple repeats of two peptides that form two orthogonal CC dimers: (S1 and S2) and (S3 and S4). Each of the polypeptide chains has a different arrangement of CC-forming repeats, either interchangeable or clustered, connected by rigid (Single-Alpha Helix, SAH); or flexible (Gly-Ser, gs) linkers. Fluorescent proteins, either TagBFP (BFP) or mCitrine (mCit) are fused to the peptides to track condensate formation and localization in cells. Due to the use of two orthogonal weakly interacting CC pairs and their different arrangement both polypeptides cannot interact in a way to satisfy all the CC interaction domains, which means that one chain can interact with multiple other chains, leading to the formation of a polypeptide network and phase separation. **b** NIH-3T3 cells transfected with 100 ng of a single plasmid. In the first image plasmid encoding protein labeled with mCitrine containing interchangeable S1 and S3 modules and linked with SAH linker. In the second image, TagBFP fused to three repeats of S2 and three repeats of S4 with a gs linker. **c** NIH-3T3 cells transfected with 100 ng of both plasmids from **b**. Cit, BFP, and merged channels are presented. Scale bars, 10 μm.

pair 1 and pair 2 are positioned interchangeably, and in the second chain they are arranged into two clusters (Fig. 1a). This strategy promotes the formation of a network with each chain interacting with several other chains. To visualize protein localization and condensate formation in living cells, proteins were genetically fused with a fluorescent protein reporter (mCit or TagBFP). Linkers between the interacting domains[55] are frequently treated as passive tethers, yet their physical characteristics (length, rigidity, charge, and sequence) may have a strong effect on phase separation by influencing binding affinity, and avidity, as well as spatial separation between interaction domains[55,58]. We initially incorporated either a long and rigid single alpha helix (SAH, composed of $E_4K_4$ repeats, 36 aa in total), or a short flexible gs linker, composed of 8 residues comprising glycine, serine, and proline, into polypeptide constructs. Polypeptides with the $E_4K_4$ pattern have a high propensity to form an elongated single alpha helix[54,56] and are thus expected to increase the physical separation between CC segments[58] in comparison to the unstructured and flexible gs linker.

HEK293 cells transfected with plasmids for different constructs showed diffuse fluorescence in the cytoplasm (Fig. 1b) or all single polypeptide chains. Rounded condensates were observed for some combinations of both polypeptide chains, suggesting that designed

CC-dimer forming polypeptides can phase separate in mammalian cells and that interactions between multiple chains are required for condensation (Fig. 1c).

## Valency influences the formation of condensates

We can control the interaction valency of each polypeptide chain through the number of CC-forming segments. Condensates were observed in polypeptide pairs where at least one of the chains had 6 CC segments and the other comprised at least 4 CC domains (Fig. 2a, b). Condensates were absent when both proteins had only 4 CC and also when one of them had 6 and the other 2 CC segments (one from each CC pair). Furthermore, we tested pairs that had matched CC arrangement in both chains but different linkers, which also led to the formation of condensates (Supplementary Fig. 3c, d). Specific interactions between CC pairs are needed, as polypeptides containing the same CC type, but with different linker types and different arrangements did not form condensates (Supplementary Fig. 4), ruling out nonspecific CC interactions.

## Liquid properties of CC-LLPS condensates in mammalian cells

Next, to confirm that the observed condensates are bona fide LLPS, we investigated the biophysical properties of condensates in

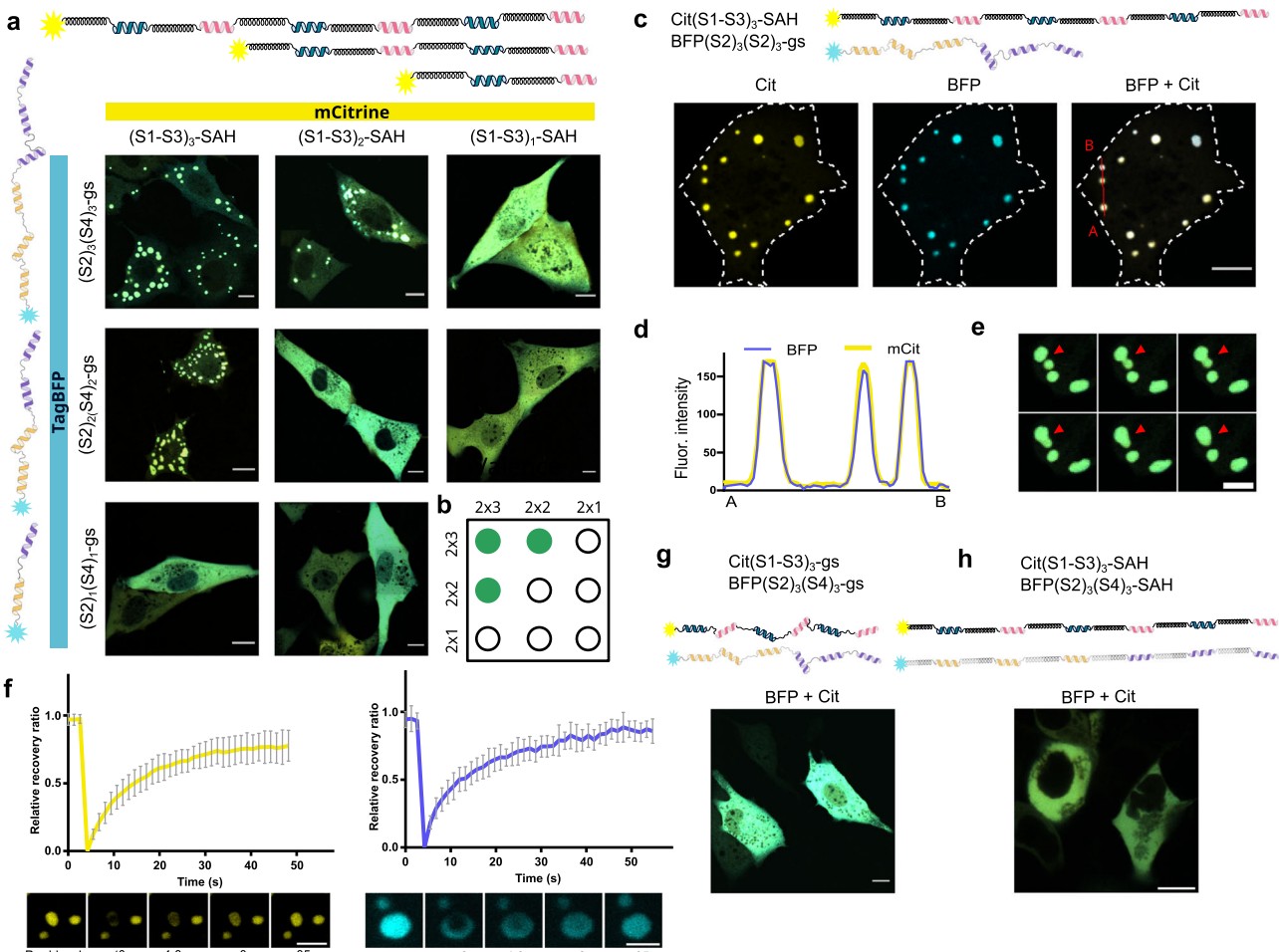

**Fig. 2 | The influence of valency and linker type on LLPS and liquid properties of CC-LLPS condensates in mammalian cells. a** NIH-3T3 cells expressing pairs of proteins with different numbers of CC repeats. From left to right, 6 CC, followed by 4, and 2 CC modules tagged with Cit. The second protein is labeled with TagBFP, with repeats of S2 and S4, interspersed with gs linker; from top to bottom 6, 4, and 2 CC segments. **b** Schematic representation of CC segments needed in each chain for the formation of condensates, green dots−condensates, empty circles−no condensates. **c** NIH-3T3 cells expressing mCit with three repeats of interchanged S1 and S3 CC, connected with SAH linkers and TagBFP with segregated three repeats of S2, and S4 CC, connected with a GS linker (same as in a) top left), separately for each channel and merged. **d** Fluorescence intensity profile showing colocalization of both fluorescent signals (mCit and BFP). Line scan of the red line in cE) from point A to B. **e** Fluorescence confocal images of condensates undergoing fusion. The red arrows indicate the locations of fusions. Insets from Supplementary Movie 1. Merge of the cyan and yellow channel. Scale bars, 10 μm. **f** FRAP analysis of the two proteins required for condensate formation. The plots show the normalized recovery after photobleaching for each protein mCit-(S1-S3)₃-SAH left yellow and TagBFP-(S2)₃(S4)₃-gs right blue graph. Data are presented as mean ± SD (*n* = 12 for mCit and *n* = 8 for BFP). Source data are provided as a Source Data file. Fluorescence confocal images of bleached condensate are presented under the plot, with the acquisition time. **g, h** Cells expressing a combination of proteins, both containing a gs linker (**g**) or a SAH linker (**h**), each with 6 CC modules in the same arrangement as a) top left. Merge of both channels. Scale bars, 10 μm.

mammalian cells. First, the colocalization of fluorescent signals from both chains (Fig. 2c, d) shows that both polypeptide chains are needed for the condensate formation. The evidence of the liquid nature of the condensates is their round appearance, minimizing surface area, as well as the ability of the droplets to fuse (Fig. 2e and Supplementary Movie 1), which was apparent in all observed rounded condensates. To evaluate the liquid nature of condensates quantitatively, we performed a FRAP analysis separately for each fluorescently labeled protein of the pair-forming condensates. Both proteins from the pair (Fig. 2c) exhibited fast recovery with halftime for mCitrine at 7.3 s and BFP at 7.4 s. Fluorescence intensity returned to more than 90 % of the initial intensity (Fig. 2f) for both fluorescent proteins, which demonstrates that the designs were indeed LLPS condensates existing in living mammalian cells.

The cellular environment may affect the formation of LLPS condensates, therefore we wanted to explore the generality of the designed system across different mammalian cell types. Formation of droplets was observed in nonadherent Jurkat T-cells and adherent HEK293T and NIH-3T3 cells, all of which are frequently used in mammalian cell research (Supplementary Fig. 5).

## The role of mismatch and CC stability on the condensate type formation

We sought to determine what role the physical properties of linkers played in LLPS formation. In our initial work, we had designed the CC proteins with SAH linkers which we hypothesized might function as a relatively rigid linker promoting the loose packing of chains required for a liquid condensate. We combined pairs containing the same linker type (Fig. 2g, h), either gs or SAH, to investigate the role of linkers. Surprisingly, when both chains contained the same linker type, either short and flexible gs or long and rigid SAH, the chains remained dispersed in the cytoplasm. Alternatively, the SAH segments, could affect the helicity of the adjacent CC-forming segments, since the helical propensity plays a prominent role in the affinity of CCs[51]. The analysis of helical propensity suggested that the CC-forming segments had substantially increased helicity when fused to an adjacent SAH-forming

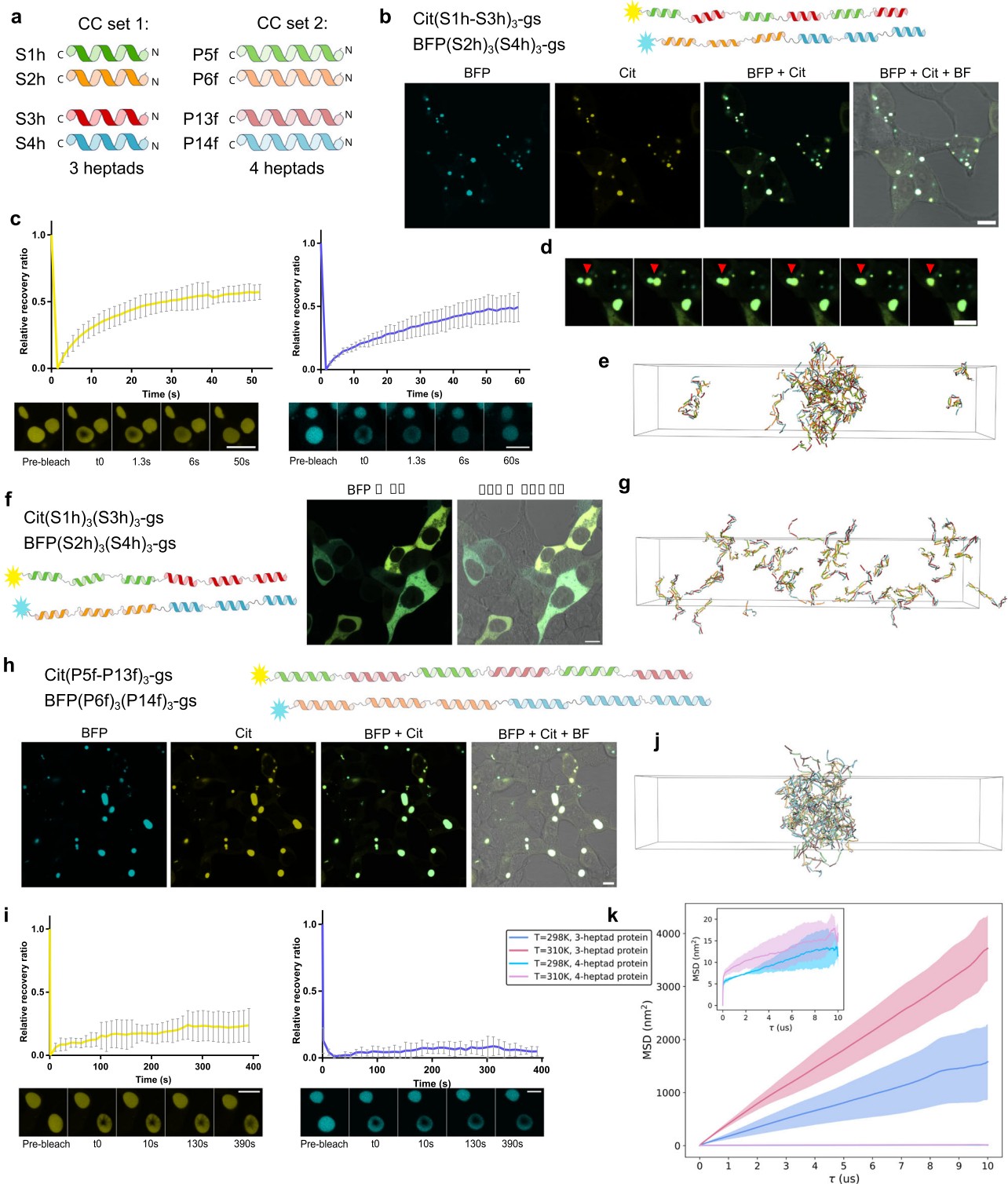

segment (Supplementary Table 3). We reasoned that increasing the helical propensity of each CC-forming segment might enable LLPS formation also with short and flexible linkers (Fig. 2g). By increasing helical propensity of CC segments −which we achieved by substituting glutamine and serine at positions *b* and *c* with alanine, which has high helical propensity[51,57] (Sh peptide set, Fig. 3a, Supplementary Table 1, and Supplementary Fig. 1) −we observed condensates even if both proteins had gs linkers (Fig. 3b). Individual chains did not form condensates (Supplementary Fig. 6). Condensates showed liquid properties with fast fluorescence recovery for both proteins (Fig. 3c), were

round in appearance, and were capable of fusion (Fig. 3d and Supplementary Movie 2).

In addition, we found that mismatch between the chains was required for LLPS formation. If the arrangement pattern of the CC segments was the same in both chains, no LLPS was observed (Fig. 3f), likely because only individual dimers formed. The mismatch between the interacting chains is, therefore, an additional parameter to hinder dimer formation and promote LLPS. Pairs of chains containing a single type of a CC dimer-forming segment did not form condensates even when the number of CC segments was increased to 12 (Supplementary

**Fig. 3 | LLPS condensate dynamics depend on the stability of CC building module interactions and can be modeled by a coarse-grained simulation.**
**a** Designed sets of CC pairs with increased helical propensity comprising three or four heptads. **b** HEK293T cells expressing polypeptide pair of two types of CC-forming segments with three heptads, with gs linkers and CC segments arranged in a different pattern forming LLPS condensates. **c** Condensates exhibit fast fluorescence recovery in a FRAP experiment left yellow for Cit and right blue for BFP channel. Data are presented as mean ± SD ($n = 9$ for Cit and $n = 7$ for BFP, condensates from different cells examined over 3 independent experiments). Below is a picture of a bleached condensate at different time points. **d** Condensate fusion. Insets from Supplementary Movie 2. Merge of the cyan and yellow channel. **e** Snapshot at 20 μs from simulation of protein constructs in **b**, at 310 K, indicative of LLPS condensate. **f** HEK293T cells expressing a pair of two types of CC-forming segments composed of three heptads, with gs linkers arranged in the same sequential arrangement remain diffuse and do not form condensates. **g** Snapshot at 20 μs from simulation of protein constructs in **f**, at 310 K, indicating no LLPS. **h** Pair of two types of CC-forming segments composed of four heptads, with gs linkers and different arrangement of CC modules in each chain, formed condensates in HEK293T cells, (**i**) which are however largely immobile, with slow and low fluorescence recovery. Data are presented as mean ± SD left yellow graph for Cit and right blue for BFP channel ($n = 5$ for Cit and $n = 3$ for BFP, condensates from different cells examined over at least 2 independent experiments). Below, bleached condensate at different time points. **j** Snapshot at 20 μs from the simulation of 4-heptad protein combination in **h**, at 310 K, is indicative of LLPS condensate. **k** Mean square displacement (MSD) analysis of proteins with mixed coil types, comparing 3-heptad coil proteins versus 4-heptad coil proteins. MSD from 4-heptad coil proteins is additionally presented as an inset due to the difference of two orders of magnitude in MSD between the two types of proteins. Lines represent the mean values, and the shaded regions are the standard deviation ($n = 3$ simulation replicates). Scale bars, 10 μm. FRAP source data are provided as a Source Data file.

Fig. 7). Interestingly, the incorporation of a longer gs linker (40 amino acid residues) into only one of the chains was sufficient to trigger LLPS (Supplementary Fig. 7d), likely due to the increased likelihood of engaging multiple binding partners by the mismatch between CC sticker spacing in the two different chains.

The unique advantage of designed CC dimers is that we can tune their affinity through the number of heptads[59] or through mutations that affect the affinity, which facilitates the identification of the sweet spot for LLPS formation. Initially, we presumed that 3 heptads might be a favorable choice, which was confirmed by testing sets composed of either two or four heptads. Two heptad modules, which are insufficient to form CC dimers, produced only diffuse distribution (Supplementary Fig. 8). Four-heptad modules, on the other hand, produced condensates with substantially decreased fluidity, due to stronger interactions that froze the assemblies into glass-like aggregates (Fig. 3h, i). FRAP analysis demonstrated poor rearrangement, with very slow and only partial fluorescence recovery (Fig. 3i), indicating 3 heptad CCs had the most appropriate stability for liquid condensates. The CC-LLPS platform thus enables gradual adjustment of the affinity of building modules to adjust the properties to the desired conditions, which could be further fine-tuned by point mutations, that introduce, for example, high helical propensity residues or increase the helicity through salt bridges.

## Simulations of coiled-coil proteins recapitulate CC-LLPS

While the rational design of de novo CC-LLPS experimentally identified parameters that guide the formation of liquid condensates, we wanted to establish a modeling platform to investigate the effect of diverse parameters, provide more molecular insight into the LLPS process, and facilitate the prediction of LLPS-forming designs. We focused our modeling on the regime of high helical propensity coils (experimentally, Sh- and Pf-set containing proteins), all of which use the gs-linker.

We used a coarse-grained framework developed in a previous study to simulate the LLPS properties of proteins containing CC segments[60]. We designed the framework to allow us to accurately capture features such as the length and segment organization of each protein as well as the interaction specificity between CC pairs with only a few tunable parameters. We justify a minimal parameter approach to focus on the hypothesis of CC-stabilized LLPS using only general rules of how CC domains are known to interact, rather than sequence-dependent information. We used this framework to ask if the rational design rules could be recapitulated and generalized and to develop a predictive tool for other polypeptide variations. The temperature scale used in the simulations is fit to match dimer binding equilibria of a single coil-coil interaction and only approximately reflects the temperature scale used in experiments.

Simulations of coarse-grained protein constructs shown in Fig. 3 agree well with experimental evidence of condensate forming designs (Fig. 3e, g, j and Supplementary Figs. 7 and 9). LLPS is determined in simulation using both density profiles and molecular cluster size distribution analyses (described in Supplemental Methods). Proteins with 3- or 4-heptad coils, but different arrangements of CC segments (Fig. 3b, h), demonstrate LLPS droplets in simulation (Fig. 3e, j and Supplementary Fig. 9a, b). Simulations of proteins with the same CC arrangement (3 + 3), as in (Fig. 3g), have properties similar to experiment and do not form condensates in simulations (Fig. 3f and Supplemental Fig. 9c). Inspection of the cluster size distributions shows that molecular dimers are the dominant species at both 298 and 310 K and appear approximately 50% and 75% of the equilibrated simulation time, respectively. This suggests that, due to the same arrangement of CC segments that align with each other (Fig. 3f, g), these proteins are forming LLPS-incompatible molecular dimers with no free binding sites.

Proteins with a single type of CC segment had markedly reduced propensity to LLPS compared to proteins with mixed patterns of coil segments (Supplementary Fig. 7), except when linker sizes are varied (Supplemental Fig. 7h). These observations are consistent with experimental data (Supplementary Fig. 7a–d). Analysis of cluster sizes in proteins with four to twelve CC segments (Supplemental Fig. 7i–k) shows that molecular dimers are the dominant cluster species in these simulations at 310 K and are present >60% of the equilibrated simulation time.

These data, in combination with results from Supplemental Fig. 9c, demonstrate that LLPS can be impaired by molecular dimers, the formation of which completely satisfies CC segment interactions when linker segments are identical in protein combinations. Simulations of proteins with 12 coils suggests LLPS at 298 K (Supplemental Fig. 7g, k) which however could not be observed experimentally at lower temperatures. This discrepancy between simulation and experiment is likely due to the inexact energy scaling inherent in our coarse-grained approach. Proteins with mixed linker sizes can realize LLPS despite having the same type of CC segment (Supplementary Fig. 7d, h, l), which is likely due to a mismatch of CC segment interactions that prevent the formation of a molecular dimer.

## Differences in the dynamics between 3- and 4-heptad proteins can be assessed using simulation

We leveraged the molecular level detail provided in the simulation to closely investigate the dynamics of protein droplets and show that the data are consistent with experimental FRAP data. While we cannot directly compare the simulation with FRAP due to the timescales of each method, we can correlate mean squared displacement (MSD) with experimental dynamics. Effective diffusion coefficients were calculated via MSD analysis with standard deviation by bootstrap sampling for the 3-heptad and 4-heptad proteins (Fig. 3k). At 298 K, the coefficients are $357 \pm 4 \times 10^{-9}$ cm²/s for the 3-heptad protein and two

orders of magnitude lower for the 4-heptad protein at $1.30 \pm 0.1 \times 10^{-9}$ cm²/s. At 310 K, the coefficients are $505 \pm 6 \times 10^{-9}$ cm²/s for the 3-heptad proteins and $2.0 \pm 0.1 \times 10^{-9}$ cm²/s for 4-heptad proteins. The difference in diffusion coefficients at both temperatures suggests that 4-heptad proteins are in a kinetic trap, as demonstrated experimentally by the formation of mostly immobile condensates (Fig. 3i). The dynamics behavior is also observed by visual analysis of trajectories and is consistent with the displacement analysis. These data support an interpretation that 3-heptad polypeptides are significantly more liquid-like than 4-heptad polypeptides.

Additional analyses show that the environment within the 4-heptad droplet is relatively static compared to the 3-heptad droplet. Individual 3-heptad coil segments, on average, interact for less time than 4-heptad coils (57% vs 65% of equilibrated simulation time at 310 K, respectively; Supplementary Fig. 9d). We also found that 4-heptad coils interact with at least 10-fold fewer unique coils than 3-heptad coils (Supplementary Fig. 9e). The lower number of unique partners analysis suggests that there is little to no mixing within the 4-heptad protein droplet on the timescale of our simulations. The higher coil affinity resulting from the 4-heptad proteins thus significantly inhibits the exchange of proteins in the droplet as well as the movement of proteins within the droplet.

### CC-LLPS condensates composed of a single polypeptide chain

Most previously described LLPS systems, including those described above, are formed from two components, either polypeptide alone or polypeptide and nucleic acids[61]. We reasoned that the use of alternative CC modules and their arrangement might facilitate the generation of LLPS from a single polypeptide chain. In this case, the design should hinder intramolecular interactions as well as the formation of homodimers with all satisfied binding sites while facilitating the formation of branched assemblies with multiple partners. CCs can dimerize either in a parallel or antiparallel orientation, which could be used to hinder intramolecular or dimeric matches. Aiming to design single-component condensates, we used two different homodimeric CC pairs, each comprising 3 heptads, one designed to pair in a parallel (P) and another in an antiparallel (A) orientation with gs linkers between them. They were designed in an APPAP arrangement to prevent the formation of straight dimers or staggered linear assemblies with satisfied binding valences.

In this arrangement, the formation of LLPS was not observed for the chain comprising 5 segments (Fig. 4b). However, duplicating the length to 10 segments (APPAPAPPAP) resulted in LLPS condensates (Fig. 4d). In this case, the appropriate sequential arrangement of segments was crucial, as another arrangement of the same number and type of CC-forming segments that were segregated (PPPPPPAAAA), did not lead to LLPS (Fig. 4c). The condensates formed from a single-chain protein Cit-APPAPAPPAP-gs (Fig. 4d) were round and exhibited liquid-like properties determined by FRAP (Fig. 4h), with fast fluorescence recovery. Additionally, fusion events were observed further demonstrating liquid properties (Fig. 4i and Supplementary Movie 3). The LLPS incompetent APPAP enriched to APPAPAPPAP condensates, acting as a client (Supplementary Fig. 9).

Simulations of orientation-controlled single polypeptides are consistent with experimental data and suggest molecular level differences from previously tested two-component systems. We modified the simulation framework to control the interaction orientation of individual coils (Supplementary Fig. 11). Snapshots from simulations of all three single polypeptide chains show agreement with experimental observations (Fig. 4e–g), and density profile and molecular cluster analysis are also consistent (Supplementary Fig. 12). Simulations suggest that the 5-coil protein, APPAP, forms relatively large clusters (Supplementary Fig. 12a) and might have intermediate LLPS propensity between the APPAPAPPAP and PPPPPPAAAA proteins (Supplementary Fig. 12a, b).

The fluorescence recovery of the APPAPAPPAP protein is slightly slower than two chain 3-heptad designs but still much faster than the 4-heptad designs (Fig. 3c, i, k, h, j). Effective diffusion coefficients and standard deviations were calculated by MSD analysis using bootstrap sampling for the APPAPAPPAP protein: 298 K, $14.9 \pm 1 \times 10^{-9}$ cm²/s; at 310 K, $158 \pm 10 \times 10^{-9}$ cm²/s. APPAPAPPAP diffusion constants are intermediate between the 3- and 4-heptad proteins, as is the MSD (Figs. 4j and 3i), consistent with experimental comparisons of fluorescence recovery.

### Coexisting multiple types of designed condensates and their fusion

Many natural condensates have been described in cells. Designed coexisting condensates, however, have only been achieved based on their consignment to different cellular locations[62]. Therefore, we wanted to investigate if the designed CC-LLPS are orthogonal to each other, as the coexistence of multiple designed LLPS compartments could be useful for diverse applications, e.g. each of them could contain different cargo and support different processes.

We introduced three orthologous designed LLPS condensates (APPAPAPPAP-gs; ((S1h-S3h)$_3$-gs + (S2h)$_3$(S4h)$_3$-gs; (P5f-P13f)-gs + (P6f)$_3$ (P14g)$_3$-gs, Fig. 5b and Supplementary Fig. 13), and found them to form three distinct condensates coexisting in the cytoplasm of mammalian cells. Furthermore, by incorporating one CC segment from the first orthologous condensate to a polypeptide of the other orthologous condensate, we achieved their fusion (Fig. 5c, d). This feature highlights the potential of orthogonal LLPS compartments in engineering cellular structures and processes.

### Engineering programmable assembly and disassembly of LLPS condensates

Our final goal was to engineer new properties into condensates and to provide regulation of CC-LLPS assembly or disassembly. As shown in Fig. 2A, condensates form when proteins comprise at least 4 CC segments, from two different sets. We hypothesized that if one of the chains from a pair were split into two chains, e.g. segmentation of BFP-(S2)$_3$(S4)$_3$-gs into BFP-(S2)$_3$-gs and (S4)$_3$-gs, in a way that each chain had only 3 CC segments of the same set, the split chains could no longer support condensation (Supplementary Fig. 14). To introduce chemically regulated assembly, we incorporated FKBP into the first chain and FRB domain into the second chain of the split protein. Upon the addition of rapamycin, those two domains heterodimerize and thus extend the length of the polypeptide. In cells expressing those polypeptides, condensates were absent in the presence of vehicle control (Fig. 6b top), however, upon the addition of rapamycin, condensates started to form after 20 minutes (Fig. 6b bottom, c and Supplementary Movie 4). A similar design was incorporated into the single chain condensates, where we incorporated FKBP/FRB domains on the 5 CC-segment polypeptides, previously shown to be insufficient to form LLPS (Fig. 4b). When two parts heterodimerize a 10 CC long polypeptide is formed, which can form LLPS (Fig. 4d). After the addition of DMSO, no condensates formed even 1-hour post-addition (Fig. 6d left), but when 1 μM rapamycin was added, the condensates were observed after a few minutes (Fig. 6d right, e and Supplementary Movie 5).

This type of regulation could also be applied in reverse, to dissociate the formed condensates by proteolytic cleavage of one of the condensate-forming chains triggering the rapid dissolution of the condensates. Dissolution was regulated using a chemically regulated split protease[63] where the addition of rapamycin (Fig. 6f) led to the dissolution of condensates after a few minutes (Fig. 6g).

We further demonstrate that condensate formation can be regulated in multi-component systems by intentionally creating LLPS-incompetent molecular dimers. We hypothesized that introducing a third polypeptide with a LLPS-capable pair, BFP(S2h)$_3$(S4h)$_3$-gs and

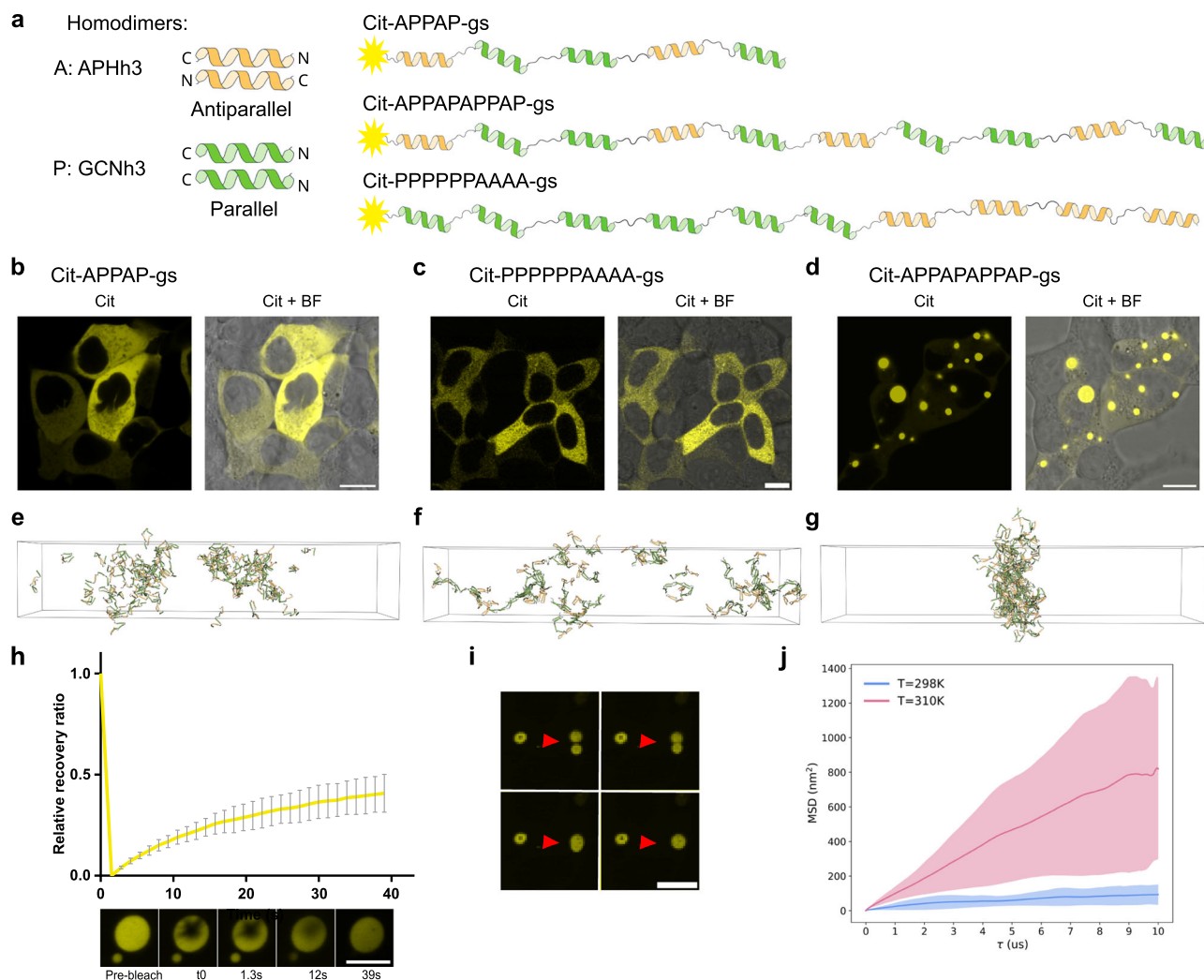

**Fig. 4 | CC-LLPS condensates from a single polypeptide chain type comprising homodimeric CC modules of the opposite pairing orientation. a** Two pairs of 3-heptad CC homodimers. APHh3 (orange) pairs in an antiparallel orientation, while GCNh3 (green) pairs in a parallel orientation. Right, representations of polypeptide chains of different lengths and arrangements of CCs connected by gs linkers. **b**–**d** HEK293T cells, expressing protein composed of mCitrine fused to different numbers and arrangements of antiparallel (A) and parallel (P) segments, with gs linkers. Scale bar, 10 μm. **b** APPAP arrangement of 5 CC, (**c**) segregated arrangement, six P followed by four As, and (**d**) APPAPAPPAP arrangement comprising 10 CC segments. **e**–**g** Snapshot at 20 μs from simulation for (**e**) APPAP arrangement, (**f**) PPPPPPAAAA and, (**g**) APPAPAPPAP arrangement at 310. **h** FRAP analysis of mCitrine-APPAPAPPAPh3 protein forming condensates. Relative recovery ratio, represented by the normalized recovery (*n* = 17 condensates from different cells examined over 3 independent experiments), of fluorescence intensity following photobleaching. The bleached region of the condensate was imaged every 1.3 s, and the plot shows the mean ± standard deviation of the normalized recovery over time. Confocal microscopy images of the bleached condensate are presented beneath the plot, with the acquisition time indicated. Scale bar, 5 μm. Source data are provided as a Source Data file. **i** Fluorescence confocal images of condensates undergoing fusion, insets from Supplementary Movie 3. The red arrows indicate the locations of fusions. Scale bar, 10 μm. **j** Mean squared displacement analysis of simulated APPAPAPPAP molecules. Lines are mean, and the shaded regions are the standard deviation (*n* = 3 simulation replicates).

Cit(S1h-S3h)₃-gs (Fig. 3b), could affect condensation. This was tested with a polypeptide that contains a cluster of S1h CC segments (Cit(S1h)₄-gs). This shorter polypeptide could pair with a cluster of complementary S2h segments on a chain BFP(S2h)₃(S4h)₃-gs to occupy its binding sites and prevent interactions with Cit(S1h-S3h)₃-gs. Indeed, as shown by simulation and experiment, this combination prevented condensation (Supplementary Fig. 15a). In contrast, a polypeptide BFP(S2h)₄-gs, which could interact with CC segments of the polypeptide (Cit(S1h-S3h)₃-gs) but has an alternating CC arrangement, did not prevent condensate formation (Supplementary Fig 15b). Furthermore, we explored an additional multi-component system featuring polypeptides with 4-heptad CCs. Cit(P5f-P13f)₃-gs was able to undergo condensation with a partner possessing only four copies of the same CC type, BFP(P6f)₄-gs. The different CC arrangement of Cit(P5f-P13f)₃-gs prevented occupation of all CC stickers in a dimer,

which, in turn, led to condensation (Supplementary Fig. 15c). In contrast, BFP(P6f)₃(P14f)₃-gs, which features a clustered arrangement, in combination with Cit(P5f)₄-gs did not result in condensate formation (Supplementary Fig. 15d), likely due to Cit(P5f)₄-gs fully satisfying the binding sites of P6f and prevented binding of additional chains to form a network. A combination of Cit(P5f-P13f)₃-gs, Cit(P5f)₄-gs, and BFP(P6f)₄-gs did not result in condensates (Supplementary Fig. 15e) due to molecular dimer formation between the two shorter polypeptides, leaving the longer polypeptide chain unable to form a network. The experimental observations—that protein combinations from b and c form condensates, whereas combinations from a, d, and e do not — were predicted by our simulation framework in a blind challenge. Density profiles, and molecular cluster distributions in particular, show simulated LLPS behavior consistent with experiment (Supplementary Fig. 15a–e, right-most graphs). The ability of our simulations to

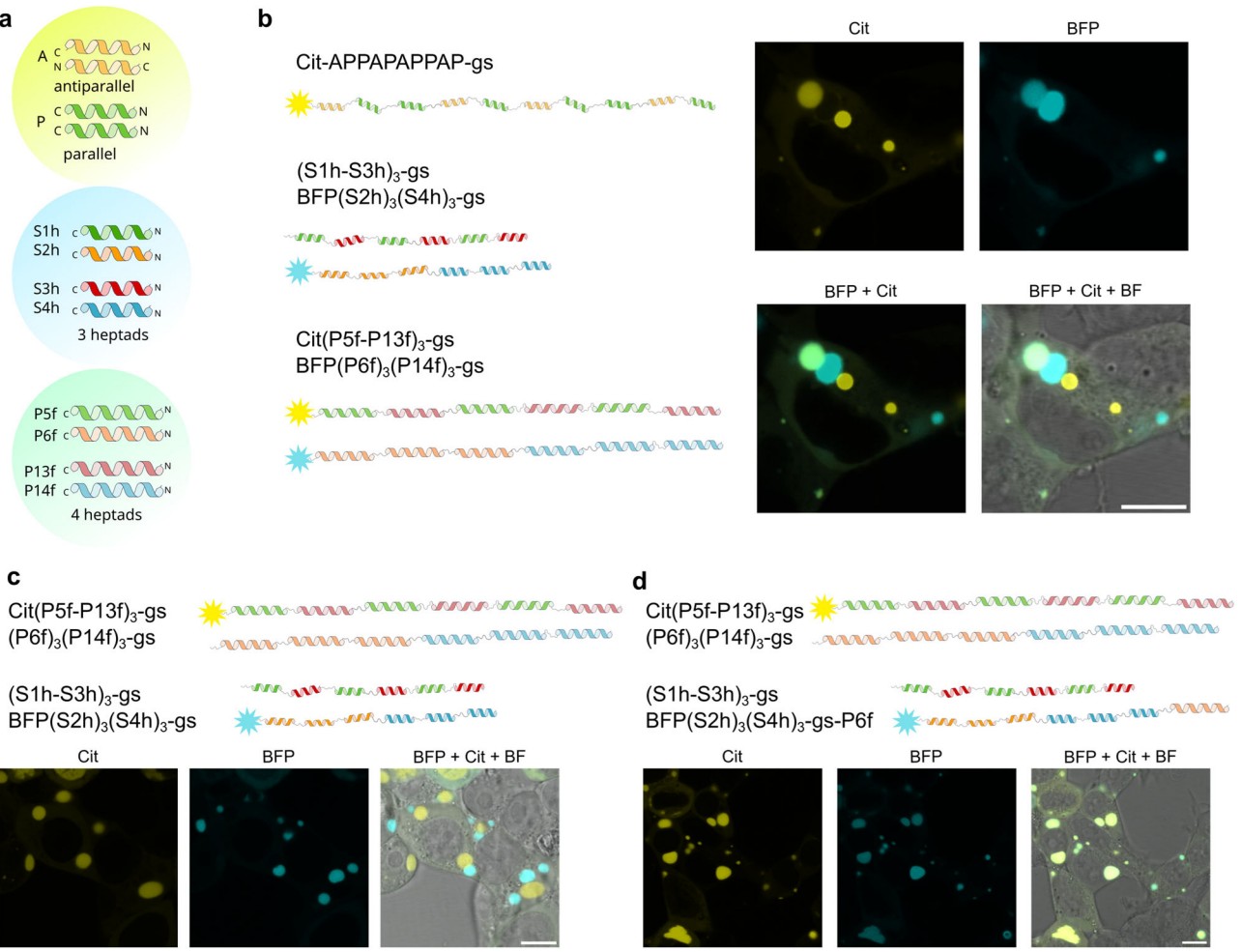

**Fig. 5 | Orthologous designed condensates coexisting in mammalian cells.**
**a** Scheme of building modules of three types of condensates, composed of 1) parallel and antiparallel modules labeled with mCitrine, 2) 3-heptad CC modules with one of them labeled with BFP and 3) 4-heptad CC modules, labeled with both Citrine and BFP. **b** HEK293T cells expressing polypeptide chains for the formation of three types of condensates. **c** HEK293T cells expressing chains that form two types of orthologous condensates. **d** HEK293T cells where one of the polypeptides from the first orthologous condensates from above contains a single CC segment (Pf6) from the second orthologous condensate, leading to mixed condensates.

robustly predict qualitative LLPS behavior demonstrates the utility of our modeling framework to enable easier protein screening for future work and helps confirm the utility of the conceptual framework used in design.

## Discussion

A deep understanding of a process can be claimed when it can be re-designed from well-understood components and processes, which also provides a mechanistic insight and tools to introduce features that may not have been observed in natural components. Protein-based LLPS formation can be achieved using diverse weakly interacting domains, such as those based on IDRs[20,35,64–68], where their nature is not entirely controllable due to transient and ill-defined interactions. LLPS formation has also been generated from a combination of naturally folded protein oligomerization domains[49,69,70], where however weak interactions cannot be excluded due to the presence of IDR segments. CC dimers present minimal binary interaction modules whose ortho-gonality can be designed, and affinity tuned in a wide range. CC-domains have been implicated in the formation of LLPS in several natural systems[4,71–76], often indirectly and in combination with IDRs, which obscured the evaluation of the contribution and the mechanism of CCs in LLPS formation.

The creation of LLPS using de novo designed CC-dimer-based proteins enables identification of the tenets of polypeptide-based LLPS, as well as dissection of the contribution of several parameters such as the affinity, valency, arrangement of interacting domains, and lin-kers. Upon identification of the minimal requirements, our research shows we can program new features into the LLPS, construct ortho-gonal condensates, and construct single-chain condensates that can be chemically regulated.

LLPS from designed CC domains occur in the absence of aromatic residues, which have been suggested along with charge segregation to play a key role in LLPS between IDPs and nucleic acids, as well as in designed polypeptide condensates[32,35,77–80]. The designed CC-LLPS confirms that the formation of condensates can be achieved through several different molecular motifs that fulfil weak multivalent inter-actions. While the de novo design of LLPS condensates from con-nected CC-forming dimers was demonstrated in mammalian cells, however, they could likely form also in other biological or biomimetic systems.

We demonstrated the key role of the interaction strength of building modules: three heptad CCs generated liquid condensates; further weakening the affinity by a truncation to two heptads gener-ated diffuse distribution, while stronger four heptad modules led to

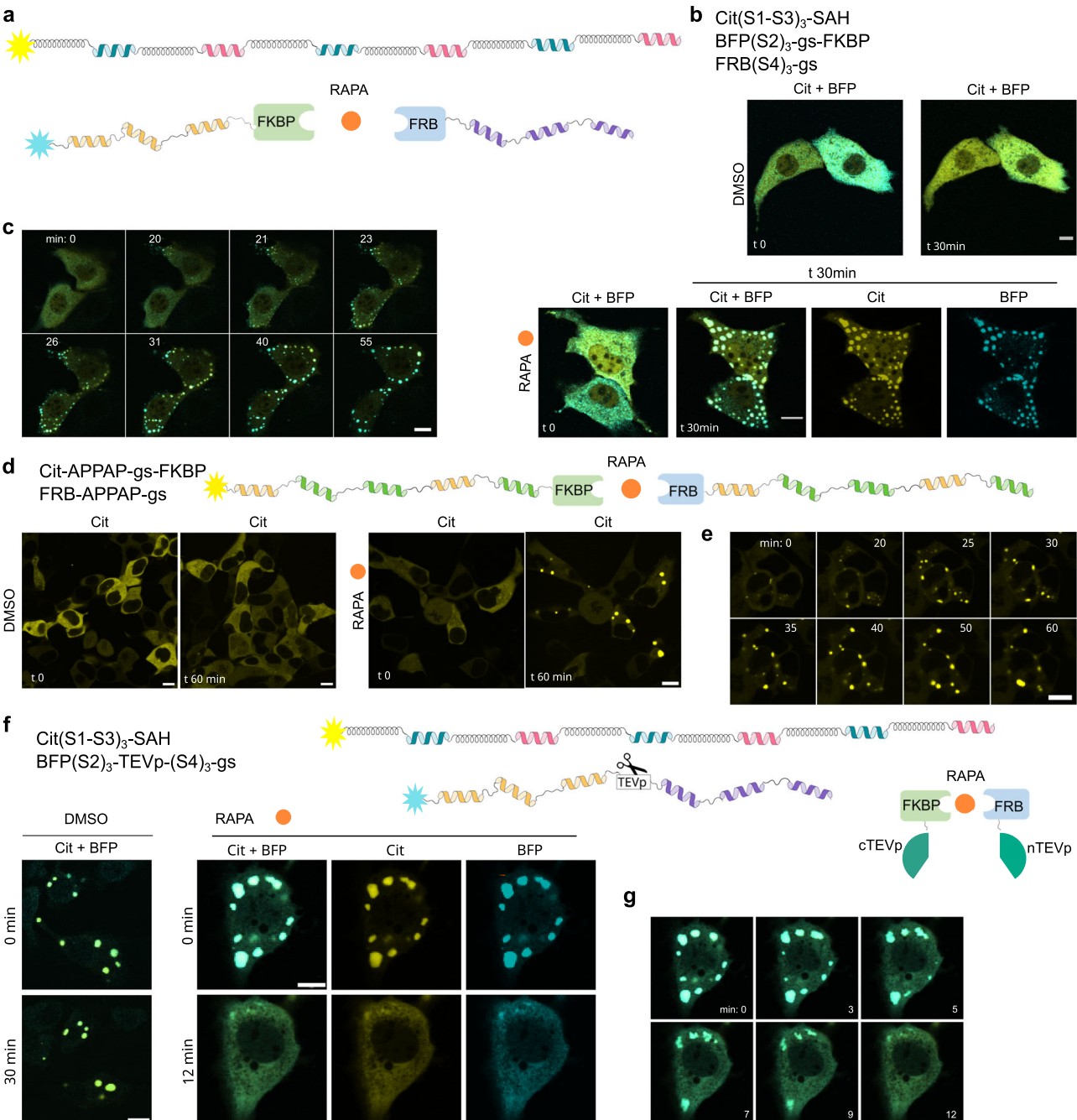

**Fig. 6 | Design of chemical regulation of CC-LLPS. a** One polypeptide chain of a condensate-forming pair BFP-(S2)$_3$(S4)$_3$-gs, was separated into two chains, each with 3 CC domains BFP-(S2)$_3$-gs and (S4)$_3$-gs. FKBP and FRB domains were fused to each chain to trigger heterodimerization by the addition of rapamycin (RAPA). **b** NIH-3T3 cells transfected with the single long chain of the pair and two chemically regulated shorter chains. DMSO was added as control and below after the addition of 1 µM rapamycin, at 0 and after 30 minutes. **c** Time series after 1 µM rapamycin addition. Merge of mCitrine and BFP channel. **d** Single polypeptide chain was split into two condensation-incompetent chains and each of them connected to either FKBP or FRB domain. HEK293T cells were transfected with both parts. Condensates were observed after the addition of rapamycin. **e** Time series after 1 µM rapamycin addition. Insets from Supplementary Movie 5. **f** TEVp site was incorporated in one of the pairs of condensate-forming proteins at a position that forms condensate incompetent pairs after cleavage. Rapamycin-regulated split TEVp protease was used. After the addition of rapamycin FKBP/FRB heterodimerize, activating split TEVp which cuts the TEVp site leading to the dissolution of condensates. HEK293T cells transfected with plasmids for both polypeptides and split TEVp. After the addition of rapamycin condensates dissolve in a matter of minutes. **g** Time series after 1 µM rapamycin addition. Scale bars, 10 µm.

the formation of immobile aggregates. The tunability of CC interaction via the selection of residues at the interacting a, d, e, g sites, helical propensity, salt bridges, and length variation facilitate manipulation of LLPS propensity. The number and arrangement of interaction segments, as well as their orientation and (mis)matching of linkers, also play an important role. Interestingly, we demonstrated that the different arrangements of the same CC-forming modules can strongly affect the formation of LLPS. Our results demonstrate that it is important to avoid the formation of dimers or small oligomers with satisfied interacting domains in a dimer. We show that this could be achieved through mismatched linkers and different arrangements or orientations of CC segments but also by an additional polypeptide

chain. CC dimers were used in this study, but it is likely that other CC oligomers, such as trimers or tetramers, could also generate CC-LLPS condensates, as suggested by simulations performed by some of the authors[60] and design in bacteria[81].

Coarse-grained simulation of CC-based LLPS with a small number of adjustable parameters demonstrated good agreement with experimental data and proved predictive in blind tests. The consistent matching of experimental and simulation data highlights the robustness of the framework we used but also demonstrates the power of the modeling approach to be extended and modified to more accurately represent CC protein interactions in simulation. Complementing the experiment with simulation also allowed us to investigate the putative behavior of individual protein constructs within a droplet, something challenging to do in vitro, let alone in vivo, to better understand how the presented design rules impact LLPS propensity. This framework can potentially be used not only to study the molecular details of LLPS-forming CC proteins but also to predict the behavior of new protein designs to streamline the engineering process.

The CC-LLPS platform (Fig. 7) demonstrated several interesting features, such as the ability to design multiple orthogonal condensates in the same cell, the formation of condensates from a single or pair of modules, and the ability to regulate assembly or disassembly by chemical regulation or additional components in a three chain system. Designed CCs did not affect natural processes in mammalian cells[42] and expression of LLPS-forming CCs also did not seem to have strong adverse effects on mammalian cells, suggesting that they might be used as a tool to modulate biological structure or processes, either for therapeutic, biotechnological, or research purposes. Overall, the well-understood structure and interaction rules of coiled coils, combined with their small size and the ability to form orthogonal pairs, make them a powerful tool in the design of synthetic membraneless condensates and for the emerging field of condensate engineering.

## Methods
### Plasmid construction
The coding DNA sequences CC repeats separated with linkers were codon-optimized for expression in human cells and synthesized by Genewiz, Leipzig, Germany GmbH, Twist Bioscience HQ, South San Francisco, California, USA, or Integrated DNA Technologies, Inc., Coralville, Iowa, USA. Fluorescent proteins BFP (TagBFP), Cit (mCitrine), mRFP and FKBP/FRB (iDimerize™, Takara Bio USA) domains were PCR amplified using repliQa HiFi ToughMix® (Quantabio, Beverly, MA, USA). Plasmids were constructed using the Gibson assembly method[82]. The amino acid sequences of all constructs are provided in Supplementary Data 1 and oligonucleotide sequences for PCR and Gibson assembly in Supplementary Data 2.

### Cell culture
The human embryonic kidney (HEK) 293 T cell line (female) (CRL-3216) was cultured in DMEM medium (Invitrogen), supplemented with 10% fetal bovine serum (FBS; BioWhittaker, Walkersville, MD, USA), the NIH-3T3 mouse fibroblast cell line (male) (CRL-1658) was cultured in DMEM-F12 medium (Invitrogen), supplemented with 5% FBS and Jurkat human T lymphocyte cell line E6.1 (male) (TIB-152) was cultured in Roswell Park Memorial Institute 1640 Medium, GlutaMAX™ Supplement (RPMI 1640 medium, Invitrogen) supplemented with 10% FBS. All the cell lines were grown at 37 °C in a 5% CO$_2$ environment. Cell lines were obtained from the ATCC culture collection.

### Transfection
NIH-3T3 or HEK293T cells were seeded at a concentration of $5 \times 10^4$ cells per well in eight-well tissue culture chambers (m-Slide 8 well, Ibidi). At ~70% confluence cells were transfected with a mixture of DNA and PEI (12 µl/500 ng DNA, stock concentration 0.324 mg/ml, pH 7.5). The amounts of transfected plasmids for microscopy were 100 ng of each plasmid, if not indicated otherwise in the figures and figure captions, to the total plasmid amount of 250 ng/well. Jurkat cells were electroporated with the Neon Transfection system (Thermo Fisher Scientific) at 1350 V, 10 ms, 3 pulses in R buffer as per the manufacturer's protocol. A total of 10–15 µg of DNA was used to electroporate $2 \times 10^6$ cells for each sample. Cells were electroporated at $2 \times 10^7$ cells per ml in 100 µl electroporation buffer and seeded in 2 ml of media. The total amount of DNA for each transfection was kept constant by adding appropriate amounts of control plasmid. Transiently transfected cells were typically imaged 48 h post-transfection.

### Chemical inducers
Rapamycin (Sigma-Aldrich) was dissolved in DMSO at concentrations of 1 mM. Before stimulation, stock concentrations of rapamycin were diluted in DMEM at 50 µM and 5 µL of the diluted inducer molecules

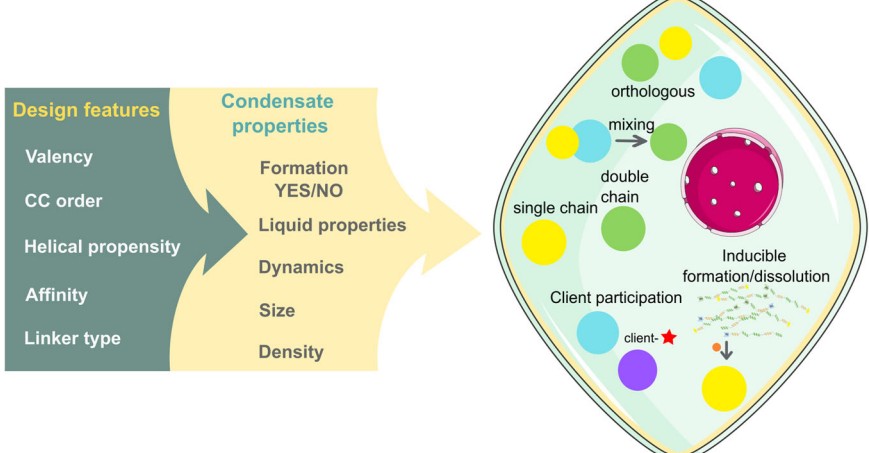

**Fig. 7 | Design features, LLPS properties and functionality of the CC-LLPS platform.** Designed orthogonal attenuated-affinity coiled-coil dimer modules can drive polypeptide liquid phase separation and condensate formation in mammalian cells. The condensate formation is modulated through the properties of coiled-coil modules such as the length, helical propensity, and sequence, defining the pairwise affinity and condensate orthogonality as well as the number of modules in each chain (valency), linker type and the arrangement (mismatch) of modules. Those parameters can influence the dynamics, size, density, and fluidity of the resulting condensates, from soluble to liquid and immobile condensates. By exploiting the adjustable parameters of the CC-LLPS system, different features, such as chemical regulation of condensate formation and dissolution, client protein incorporation, and orthogonal coexistence of multiple condensates can be introduced.

were added to a well in an 8-well chamber for a final concentration of 1 μM or control DMSO diluted in the same ratio was added, directly before imaging.

## Microscopy

Cells were imaged two days after transfection with Leica TCS SP5 inverted laser-scanning microscope on a Leica DMI 6000 CS module, using x63 oil-immersion objective, numerical aperture (NA) 1.4 (Leica Microsystems). For fluorescence imaging, a 514-nm laser line of a 100-mW argon laser with 30% laser power was used for mCitrine excitation, and the emitted light was detected between 530 and 550 nm. For TagBFP excitation a 50-mW 405-nm diode laser was used and the emitted light was detected between 420 nm and 460 nm. A 10- mW 543-nm HeNe laser was used for mRFP excitation, and the emitted light was detected between 600 nm and 640 nm. Further, Leica LAS AF software was used for acquisition, images were processed with Fiji software.

## Fluorescence recovery after photobleaching

Fluorescence recovery after photobleaching (FRAP) experiments on condensates in live NIH 3T3 or HEK293T cells were performed starting approximately 48 h after transfection. A cell or part of a cell containing condensates was first scanned to establish the initial fluorescence and a part of a single condensate was bleached with a 488-nm laser or 405-nm laser line for mCitrine and TagBFP, respectively. Fluorescence recovery was monitored by image acquisition every 1,3 s or every 5 s for BFP(SAH-S2)3(SAH-S4)3. For each experiment, the fluorescence intensities of a neighboring droplet with similar size to the bleached one were also recorded to correct for photobleaching. Raw data were background subtracted and normalized using Excel and plotted using GraphPad Prism 6.01 software. Data are presented as the average ± SD. Fitting was done in GraphPad Prism by using non-linear regression and the exponential one-phase association model using Y0 = 0, from which the halftime recovery and the plateau were obtained.

## Circular dichroism spectrometry

Synthetic peptides (Proteogenix) were dissolved in in Tris buffer (50 mM Tris pH 7.5, 150 mM NaCl, 1 mM TCEP) with a protein concentration between 0.5 mg/ml and 2 mg/ml, peptide concentrations were determined using Pierce™ BCA Protein Assay Kit (Thermo Fisher Scientific) per manufacturer protocol. The experiments were conducted on a Chirascan CD spectrometer (Applied Photophysics) equipped with a Peltier temperature controller. The individual peptide concentrations were maintained at 40 μM, and the peptide mixture consisted of 20 μM for each peptide in a Tris buffer. Quartz cuvette (Hellma, Germany), with 1 mm path length was used in all the experiments. CD spectra were collected at 1 nm intervals over the wavelength range of 200 to 280 nm, with a 1 nm bandwidth and an integration time of 0.5 seconds. Unless otherwise specified, the reported results represent the average of six scans at a temperature of 25 °C.

## Isothermal titration calorimetry

MicroCal PEAQ-ITC (Malvern) was used to measure the equilibrium dissociation constant (Kd) of orthogonal CC pairs. A 40-μl injection syringe was filled with the peptide solution (1 mM P5f; P13f; 5.6 mM S1h or 5.4 mM S3h) in 50 mM Tris pH of 7.5, 150 mM NaCl, and 1 mM TCEP. The samples were injected into an ITC cell (volume 200 μl) filled with a complementary peptide (100 μM P6f or P14f and 560 μM S2h or 540 μM S4h) in a matching buffer through a 19-step process. The injection volumes were 2 μl and were spaced 120 s apart, except for the first priming injection of 0.4 μl with 150 s spacing. Raw thermograms were integrated with the software NITPIC[83], interaction analysis was done with SEDPHAT[84], and titration curves were visualized with a companion GUSSI software. Monte Carlo simulation for non-linear regression, aligning the model function with experimental values 1000 times was used to obtain uncertainties in the fitted model.

## Simulation framework for coiled-coil LLPS

We utilized the recently developed CC LLPS framework[60] to simulate the phase separation of models of the Sh- and Pf-coiled-coil-containing proteins, and the parallel/antiparallel single polypeptide constructs. This framework represents CC proteins as assemblies of coil segments (i.e. CC domains) and flexible linker segments (i.e. tethers between CC domains), using C-α coarse-graining. General information about the framework and its assumptions can be found elsewhere[79]. We use the same interaction terms in simulations for this study as in the original CC LLPS framework, with the exception that sticky interactions between coil segments use σ = 0.570, and ε = 7.5 kJ/mol. These parameters were changed so that the shorter CCs in this study could form dimers with similar strength as the CCs used in the original CC LLPS framework. We also generated an orientation-specific version of the framework to allow for control over parallel versus antiparallel CC interactions for simulations of the single polypeptide constructs. Details about the interaction strengths specific to this study, and the orientation-specific framework, are described in Supplemental Methods. Fluorescent protein tags, which are used experimentally, are not included in our coarse-grained protein models.

## Simulations to assess phase coexistence

We used the slab simulation protocol[60,79] to directly simulate the phase coexistence of CC protein models. The slab protocol consists of the following workflow: run single molecule simulations of individual proteins, pack a simulation box with configurations of a protein (or protein pair for Sh- and Pf-CC proteins) from the single molecule molecular dynamics (MD), energy minimize the packed box, compress the box in the z-dimension to make the slab (NPT ensemble, 150 ns), expand the box in the same compressed dimension to ~10 times the compressed length, then perform production simulations (NVT ensemble, 20+ μs). We performed single molecule MD and slab simulations at 298 and 310 K. MD parameters, including integrators, cutoffs, etc., are the same as in ref. 60. We analyzed the molecule number density (density profiles) and molecular cluster sizes to determine LLPS. Simulations with a density transition and molecular clusters containing nearly all the proteins in the simulation are markers of LLPS. We additionally quantified the diffusion of select protein pairs from mean squared displacement analysis. Further details are provided in Supplemental Methods.

## Statistics and reproducibility

All microscopic images are representative images of at least three independent experiments and five separate observations within the same experiment. The density profile, molecular cluster, mean squared displacement, and coil interaction analyses are from three simulation replicates. Snapshots are representative of simulations of three independent replicates.

## Software

Molecular dynamics simulations were performed using unmodified GROMACS v2022.1[78], and custom analysis code was written in Python 3. 9 (link below). Compilation details are the same as in[60]. Graphs and statistical analyses were prepared with GraphPad Prism.8.4.3 (http://www.graphpad.com/). The microscopy pictures were analyzed and prepared with LAS X, Leica (https://www.leica-microsystems.com/products/microscope-software/p/leica-las-x-ls/) and ImageJ (https://imagej.nih.gov/ij/). Schemes were prepared with Inkscape™ 0.92.3, Brooklyn, NY (https://inkscape.org/).

## Reporting summary

Further information on research design is available in the Nature Portfolio Reporting Summary linked to this article.

## Data availability

The data generated in this study and supporting the findings of this study are available in the paper and its Supplementary Information files. Source data are provided with this paper. Any other relevant data or reagents are available from the corresponding author upon request without limitation. In case of interest for DNA constructs they are available via Addgene. Source data are provided with this paper.

## Code availability

GROMACS topology files and MD parameter files, as well as key analysis codes, are available on the GitHub repository: https://github.com/dora1300/programmable_coiledcoil_llps and Zenodo https://doi.org/10.5281/zenodo.10080721[85].

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

## Acknowledgements

We thank Tadej Satler for help with ITC data analysis. This work was supported by grants from the Slovenian Research Agency (P4-0176, P3-0289, J7-4640, J1-4406). This work is part of a project MaCChines that has received funding from the European Research Council (ERC) MaCChines under the European Union's Horizon 2018 research and innovation program Grant agreement ID: 899259 (R.J.). The Alpine high-performance computing resource at the University of Colorado Boulder is jointly funded by the University of Colorado Boulder, the University of Colorado Anschutz, Colorado State University, and the National Science Foundation (award 2201538). This work was supported by the NIH Molecular Biophysics Training Program (T32GM065103, D.R.), and NSF 1943488 (L.H.). In Fig. 7 emptycell-oval icon by Servier https://smart.servier.com/ is licensed under CC-BY 3.0 Unported.

## Author contributions

R.J. conceived and supervised the study. M.R. and R.J. designed the experiments. M.R. performed the experimental work, K.E.F., G.A., and S.V. helped in determination of the affinity of CCs, D. A. R. performed simulations, M.R.S, L.H., and D.A.R conceived, designed, and interpreted the simulations, and R.J. and M.R. analyzed and discussed the experimental results. All authors discussed and commented on the manuscript before submission.

## Competing interests

M.R.S. is a Open Software Fellow with Psivant Therapeutics, and consults for Relay Therapeutics. The remaining authors declare no competing interests.
