## [Peer Review File · Nature Communications]

Reviewers' Comments:

Reviewer #1:

Remarks to the Author:

In this manuscript, Ramsak et al developed biomolecular condensates that are formed by engineered coiled-coil (CC) polypeptides in mammalian cells. By using CC polypeptides with varying length and compositions, including linkers with different helical propensity, the authors could both gain insights into the role of polypeptide composition and dimer formation in LLPS, and regulate the resulting condensate properties. This is a very thorough and extensive work, and especially impressive from the polypeptide design perspective. The authors studied how ordered vs. disordered linkers, number of CC segments, their composition, and different combinations of CC pairs affect LLPS. Specifically, the authors show how the strength of interaction between the pairs affect the condensate dynamics (faster vs. slower diffusion). By these systematic studies, and with the aid of simulations, the authors concluded that one critical condition for LLPS is that the polypeptide should not fully form CC dimers, and that the latter can be engineered by utilizing different mismatched linkers. This system can be further developed in vitro to further gain molecular level insights into LLPS of CC polypeptides or in cells, for biotechnological or biomedical applications. I believe that the extensive analyses and insights shown in the manuscript will be of great interest to the field, especially to the growing community of researcher who focus on engineered condensates, and more broadly to the peptide/protein self-assembly and LLPS communities. Therefore, I highly recommend publishing the work after addressing the few points detailed below.

1. The design principles of the CC dimers for LLPS are the in the core of this manuscript and in my opinion should be better explained in the main text (rather than in the Methods, as in the current version). For example, how the authors managed to design loosely interacting CC dimers, and which segments interact with which.
2. Please indicate clearly what is the length or MW of each of the protein segments used in the manuscript using a table or alike.
3. Page 4 line 103: "likely due to the decreased branching propensity". Please explain.
4. Do the authors have any experimental evidence for the helical content of the proteins with SAH vs. gs linkers?
5. Figure 2 panel D: the authors concluded that when both CC pairs contain either gs or SAH linkers, they remained dispersed in the cytoplasm. However, from the confocal microscopy image in Figure 2d, it looks like the proteins containing the helical linker SAH assemble into large structures or aggregates in the cytoplasm – please provide more details and clarify if these proteins indeed aggregate.
6. FRAP experiment: the authors indicate that the halftime of the proteins is about 8 sec yet from the recovery plots in Figure 2 it seems that the $t_{1/2}$ is larger than 10 sec. Please provide accurate values and provide details on how this parameter was calculated.
7. A follow-up question to the previous one: it is a bit surprising that mCit-(S1-S3)3-SAH and TagBFP-(S2)3(S4)3-gs have similar $t_{1/2}$ considering their different size. Please discuss this.
8. Since this is a cellular expression system, the authors are limited in their ability to characterize the polypeptide building blocks and thus cannot determine the % of helical vs. unfolded content for each of the polypeptide i.e., using FTIR or CD. Are the authors plan to study LLPS of these CC polypeptides in vitro, in the future? It might be worth adding a comment on this in the manuscript.
9. How do the authors explain the correlation between increased helicity and increased LLPS propensity? This is in contrast with other biological condensates in which disordered regions within protein are the main interacting segments. In general, I think that the authors should think of a potential hypothesis about the effect of intramolecular order/disorder (helicity vs unfolded regions) in their system on cellular LLPS.

Minor comments:

- Figure 1 is extremely complicated and challenging to follow. Please think of a way to simplify it, explain in more details in the text about each peptide and avoid abbreviations as much as possible.
- For clarity, page 5 lines 133-135 should be moved to the paragraph on page 6 which focuses on the effect of linkers on LLPS.
- Supplementary table 2: is the gray color code mentioned in the caption of the table supposed to

be black? As no gray color appears in the sequences.

Reviewer #2:

Remarks to the Author:

Ramšak et al. present an elegant study on the design of liquid-liquid phase separating proteins using de novo coiled coil domains. Rather than focusing on engineering existing proteins, the authors highlight the power of bottom-up protein design for self-assembly.

In this study, the authors use coiled coil dimers as modular building blocks for multivalent polypeptides that undergo LLPS. The authors empirically test different combinations of CC-forming domains while varying the number, lengths, and arrangements, as well as the rigidity and combinations of different linkers. In this way, they determine the appropriate valency and arrangement for phase separation. Perhaps uniquely in such engineered phase separating systems, the authors are able to test concepts such as mismatch between stickers in a biological setting, and rationally alter interaction affinities using coiled coil design rules. From a protein design perspective, this is perhaps the most interesting achievement of the study, as it demonstrates the capacity for the system to be understood in a way that most phase separating proteins cannot. Further, the authors use molecular dynamics to simulate the changes made to the designed molecules. Finally, the authors show that the designed molecules can be made orthogonal, and assembled and disassembled with previously well-characterised chemically inducible protein-protein interaction motifs (FKBP/FRB).

Overall, the study shows the clear and growing role of protein design in LLPS, and we can only applaud their success in delivering de novo proteins forming highly dynamic liquid droplets in mammalian cells. The authors have developed some unique systems with their designs but have not used in vitro experiments at all to verify either their hypotheses around the coiled coil propensity or LLPS. We would encourage the authors to develop their study further, and make it a really strong contribution to the field, which could then be fully endorsed for publication in Nature Communications.

1. The authors refer to established coiled coil dimer design rules for their dimer designs, however there is little detail on the selection of the specific sequences used (i.e., where did they come from, and why were these sequences chosen?). Can the authors clarify the design choices here?
2. Further, the authors have made hypotheses regarding coiled coil propensity and orthogonality and its impact on LLPS, but these hypotheses have not been tested in vitro by standard biophysical methods. This is particularly poignant as the authors note the importance of weak interactions for LLPS, but they have not characterised the strength and specificity of the dimers they used. This would be very interesting for informing the relative strengths of PPIs for phase separation, which is challenging to interrogate in other systems. Can the authors please provide biophysical characterisation of the utilised CC building blocks to support their rationalisation of interaction strength and valency in their designs.
3. Why do the designer proteins require two orthogonal types of coiled coil dimers? The authors claim this is due to decreased branch propensity, but could it imply some crosstalk between supposedly orthogonal motifs? Is it pointing to the importance of promiscuous interactions of amphipathic helices rather than well-defined coiled coil dimers? Further, the reciprocal control for Fig. S1 (Cit(S1)6-SAH and BFP(S3)6-gs) is missing.
4. S1-4h peptides are described as being designed by introduction of intrachain salt bridges. However, based on Table S1 they appear to differ from peptides S1-4 only by replacement of the b and c QS/SQ residues with AA. This does not appear to introduce additional intrachain salt bridges. Can the authors please confirm and clarify their design process here? Further, the authors seem to ignore a significant change in the protein hydrophobicity and H-bonding capacity caused by the QS->AA mutations that could be important for PPI and LLPS.
5. Fig. 2D seems to be contradicting the text. The constructs with SAH linkers do appear to form some kind of condensates. Could the authors review these data, and provide some rationalisation for these results?
6. The molecular dynamics simulations used are interesting, and their relation to experiment has been performed well. However, we are not convinced that the simulations are acting predictively, which would be the ultimate goal of such a tool. Can the authors please provide some evidence that their simulations can behave predictively? For instance, the authors note that simulations of proteins with 12 coils suggested LLPS at a lower temperature. The experimental verification of this

(or a similar in silico led hypothesis) would be extremely satisfying, and demonstrate that the MD simulations are able to act predictively.

7. The use of FKBP/FRB as a chemically inducible heterodimeric PPI was perhaps prudent, but this system has already seen extensive use as a chemical inducer of, or recruiter to, protein condensates, and so it potentially does not bring a great deal of novelty to the study. As the rest of the study has covered the systematic design of coiled coils and their arrangement so comprehensively, this final section is somewhat underwhelming. The authors have made a very interesting orthogonal multi-component system with their coiled coil dimers, and it would potentially have been nice to see this applied to something instead to prove the utility of this system.

Technical comments:

1. Reference 53 does not appear to direct to the appropriate literature (description of a coarse-grained framework, line 203 of the manuscript PDF). Could the authors please confirm that the correct literature is referenced here?

2. Overall, while the design framework used is overall comprehensive, it is occasionally challenging to follow in the manuscript and could benefit from restructuring for clarity. Mentioning the concept of "stickers and spacers" (which these designs clearly resemble) would also be beneficial as it has been widely used to rationalise LLPS of proteins. A summary table of designs in the supplementary information could help direct the reader.

Reviewer #3:

None

REVIEWER COMMENTS

Reviewer #1 (Remarks to the Author):

In this manuscript, Ramšak et al developed biomolecular condensates that are formed by engineered coiled-coil (CC) polypeptides in mammalian cells. By using CC polypeptides with varying length and compositions, including linkers with different helical propensity, the authors could both gain insights into the role of polypeptide composition and dimer formation in LLPS, and regulate the resulting condensate properties. This is a very thorough and extensive work, and especially impressive from the polypeptide design perspective. The authors studied how ordered vs. disordered linkers, number of CC segments, their composition, and different combinations of CC pairs affect LLPS. Specifically, the authors show how the strength of interaction between the pairs affect the condensate dynamics (faster vs. slower diffusion). By these systematic studies, and with the aid of simulations, the authors concluded that one critical condition for LLPS is that the polypeptide should not fully form CC dimers, and that the latter can be engineered by utilizing different mismatched linkers. This system can be further developed in vitro to further gain molecular level insights into LLPS of CC polypeptides or in cells, for biotechnological or biomedical applications. I believe that the extensive analyses and insights shown in the manuscript will be of great interest to the field, especially to the growing community of researcher who focus on engineered condensates, and more broadly to the peptide/protein self-assembly and LLPS communities. Therefore, I highly recommend publishing the work after addressing the few points detailed below.

We thank the reviewer for their supportive and insightful comments. We address the reviewer`s specific comments below:

1. The design principles of the CC dimers for LLPS are the in the core of this manuscript and in my opinion should be better explained in the main text (rather than in the Methods, as in the current version). For example, how the authors managed to design loosely interacting CC dimers, and which segments interact with which.

We agree that the reviewer made a valid point and have thus moved the CC design principles from the Supplementary information to the main text as well tried to explain the principles of the design in more detail as well as included clearer information about the designed and used CC pairs: 'We designed an orthologous set of weakly interacting CC pairs, with adjustable affinities suitable for LLPS formation (sequences in Supplementary Table 1). The design was based on the same principles as used previously for the design of strongly interacting orthogonal pairs¹⁻⁴. Canonical parallel CC dimers are composed of heptad repeats, with positions labeled 'abcdefg'.

'Our initial set of CC segments, labeled 'S', has the weakest affinity with 3-heptad CC modules (S1-S4), where S1 was designed to form a pair with S2 and S3 with S4. At positions *b, c* we introduced amino acid residues Gln and Ser, to weaken the helical propensity...'

2. Please indicate clearly what is the length or MW of each of the protein segments used in the manuscript using a table or alike.

To improve on the clarity of our study, we have provided a comprehensive table of the constructs used, accompanied by their corresponding molecular weights in the supplementary material - Supplementary Table 2.

3. Page 4 line 103: “likely due to the decreased branching propensity”. Please explain.

We replaced the term branching with the formation of a network. For condensates to form, a polypeptide chain needs to be able to interact with multiple other chains rather than form a dimer or a linear assembly. This ability of a polypeptide chain to interact with multiple other chains enables the formation of a network.

We included a short explanation in the main text: “If only a single type of the CC-forming peptide was present in each of the two polypeptide chains, no condensates were formed, as they can only form dimer or linear polymers and does not enable formation of a network ”

4. Do the authors have any experimental evidence for the helical content of the proteins with SAH vs. gs linkers?

The helical content of SAH linkers was inferred only based on the prediction of its helical propensity and numerous reports on the high helical content of peptides with the E₄K₄ repeats. As described in the manuscript, the presence of SAH linkers increased the predicted helical propensity of three-heptad peptides that themselves have very low helicity and strongly increased the formation of condensates in contrast to polypeptides with gs linkers.

5. Figure 2 panel D: the authors concluded that when both CC pairs contain either gs or SAH linkers, they remained dispersed in the cytoplasm. However, from the confocal microscopy image in Figure 2d, it looks like the proteins containing the helical linker SAH assemble into large structures or aggregates in the cytoplasm – please provide more details and clarify if these proteins indeed aggregate.

We apologize for any misunderstanding caused by the presentation of this experiment. We did not observe the formation of *bona fide* liquid condensates under those conditions. The images presented in Fig. 2d gave the impression of condensed regions but this was rather due to the cellular structures. Furthermore, we want to clarify that in none of the cells we observed the presence of round and droplet-like structures, as expected from liquid condensates. Therefore, we selected for presentation other images that were more representative and we show several additional images below.

To address these concerns and provide a more accurate representation of our findings, we have included pictures of more representative cells in our revised presentation (now. Fig. 2h).

6. FRAP experiment: the authors indicate that the halftime of the proteins is about 8 sec yet from the recovery plots in Figure 2 it seems that the t_{1/2} is larger than 10 sec. Please provide accurate values and provide details on how this parameter was calculated.

To clarify our methodology, we first normalized the fluorescence intensity by subtracting the background intensity for each time point. Following this, we divided the values by the intensity of a neighboring unbleached condensate. The pre-bleach values were set as max intensity and the bleached point as minimum. Means + SD were calculated.

In order to fit the data with one-phase exponential equations, we set the bleached data point at time 0. This approach allowed us to determine the actual half-time of fluorescence recovery. This adjustment is the reason for the seemingly longer $t_{1/2}$ in the graph presented in Figure 2. The exact $t_{1/2}$ for mCit(S1-S3)₃-SAH is 7.3 s and for BFP(S2)₃(S4)₃-gs 7.4 s. Perhaps the $t_{1/2}$ seems at first glance to be longer as the time scale starts at the beginning of bleaching and not when the recovery starts.

We included the information for halftime calculation in methods, FRAP: Fitting was done in GraphPad Prism by using non-linear regression and the exponential one-phase association model using $Y_0 = 0...$

In the main text the value of $t_{1/2}$ from the fit is now included: Both proteins from the pair (Fig. 2c) exhibited fast recovery with halftime for mCitrine at 7.3 seconds and BFP at 7.4 seconds...

7. A follow-up question to the previous one: it is a bit surprising that mCit-(S1-S3)₃-SAH and TagBFP-(S2)₃(S4)₃-gs have similar $t_{1/2}$ considering their different size. Please discuss this.

If the condensates are composed of the two components then the $t_{1/2}$ would be expected to be similar for both components, due to the dynamics of the liquid condensate.

We have observed a consistent trend in our data (also in Fig. 3g) that polypeptides with a segregated arrangement of coils (3+3) in general tend to exhibit slower recovery rates than polypeptides with alternating coils, although the exact reasons behind this phenomenon remain unexplained at this stage. It is possible that the stoichiometry is different from 1:1, which could affect the average diffusion rate for each component.

The results shown below are FRAP results of a combination of polypeptides mCit(S1-S3)₃-gs and BFP(S2)₃(S4)₃-SAH, where the linker type is reversed between the two polypeptides from Figure 2c. In this case the difference is larger between the two components which could be a combination of the effect of different size and segment arrangement.

8. Since this is a cellular expression system, the authors are limited in their ability to characterize the polypeptide building blocks and thus cannot determine the % of helical vs. unfolded content for each of the polypeptide i.e., using FTIR or CD. Are the authors plan to study LLPS of these CC polypeptides in vitro, in the future? It might be worth adding a comment on this in the manuscript.

The reviewer presents an important point; we plan to study and characterize polypeptides in vitro, their biochemical properties, as well as their ability to phase separate in vitro. We have, additionally analyzed the CD spectra of individual peptides and pairs used in the LLPS-CC:

Supplementary Figure 1: Circular dichroism (CD) spectra of a 1:1 mixture of CC.

Spectra comparison of single peptide and peptide pair for a, b) 3-heptad CC designed to have weak affinity; b, c) for 3-heptad CC designed with increased helical propensity and e, f) 4-heptad CC. g, h) Spectra depicting orthogonality, showcasing the comparison between peptide pairs and non-pair peptide combinations for g) 3-heptad peptides and h) 4-heptad peptides.

The CD analysis provides compelling evidence in support of our design strategy. Specifically, we found that the S1-S4 peptides exhibit minimal helical content when analyzed individually or in pairs. However, when we examined the pairs with middle affinity (the Sh versions), we

observed a noticeable increase in their α -helical content in combination with their designed pairs. This increase is accentuated at lower temperatures.

As per our design expectations, the 4-peptide coiled-coil (the Pf version) exhibited the highest helical content. Once again, we observed a significant augmentation in helix formation when the coiled-coil pairs were present compared to when a single peptide was analyzed in isolation.

These results not only validate our initial design hypothesis but also provide valuable insights into the structural properties of these peptides. We have included the above figure in the supplementary material – Supplementary Fig. 1.

Importantly, we have analyzed the binding of peptide pairs using ITC :

Supplementary Figure 2: Binding analysis using isothermal titration calorimetry (ITC). For a) peptide pair S1h + S2h and b) S3h + S4h, which are 3-heptad CC with higher helical propensity. For peptide pair c) P5f + P6f and d) P13f + P14f, which are 4-heptad CC.

We determined that the Pf set of peptide pairs have affinities in the range of a few μ M, while Sh peptides comprising 3 heptads have much lower affinities. We could not determine the affinity of the S set of peptides using ITC and they are likely above the mM range.

We have included the above figure in the Supplementary material – Supplementary Fig. 2.

9. How do the authors explain the correlation between increased helicity and increased LLPS propensity? This is in contrast with other biological condensates in which disordered regions within protein are the main interacting segments. In general, I think that the authors should think of a potential hypothesis about the effect of intramolecular order/disorder (helicity vs unfolded regions) in their system on cellular LLPS.

Our designed systems agree well with the “stickers and spacers model”, where the stickers are CC-dimer forming segments where the increased helicity augments their affinity to the

appropriate range to form condensates yet maintains their fluidity. It has been established before that an increase in helicity corresponds to a higher affinity within coiled-coil interactions². Our system differs from most biological condensates that primarily rely on interactions within the disordered regions for their function. In case of CC-LLPS, the interactions between weakly, yet tunable, interacting domains (specifically, the coiled-coil dimers) resemble systems like SUMO/SIM⁵.

The disordered regions (linkers, spacers) enable spatial flexibility to allow the stickers to engage in multivalent interactions, and at the same time tether the stickers into the covalently linked multivalent chain. Even disordered regions in most natural condensates have to provide certain affinity, which however is not so well defined or even tunable as in case of weakly interacting CC dimers. Importantly, the disordered regions in our proteins are not the driving force behind condensation, as polypeptides lacking coiled-coil pairs do not undergo LLPS.

The reviewer might be referring to the need for flexibility that would allow a protein to adopt many different conformations that can engage in intermolecular interactions. In all of our systems that form LLPS, at least one of the pairs had flexible linkers. So, although the SAH-containing partner may be relatively rigid, the flexibility required to engage in multiple contacts (or branching propensity) is retained by the more flexible partner.

The reviewer may additionally be asking whether the helical structure of the coils contribute in some unique fashion to LLPS formation that is distinct on a molecular level from that of stickers made from disordered regions. While this is certainly of interest and part of our ongoing work, the goal of this paper is to highlight the potentials of coils in their amenability to rational and facile design. Our successes at engineering multiple distinct CC-containing proteins that drive LLPS demonstrate that *de novo* design of LLPS through CCs is very tractable. Thus, this can provide a more readily accessible platform for engineered functional LLPS than their disordered counterparts.

To provide further supporting results, we have included a new Supplementary Figure 4 in the revised Supplementary information shown below. This figure demonstrates that when both polypeptides contain the same coiled-coil segments, but different linker types, condensation does not occur. This reinforces the critical role of precisely tuned coiled-coil pair interactions in our system.

Supplementary Figure 4: Polypeptides containing the same CC type do not form condensates in cells.

a) mCitrine fused to the repeats of S1 and S3 in an interchangeable arrangement and separated with SAH linker and mCitrine fused to the repeats of S1 and S3 in an interchangeable arrangement and separated with gs linker. b) TagBFP fused to three S2 and three S4 separated with SAH linker and TagBFP fused to three S2 and three S4 separated with gs linker. c) NIH-3T3 cells expressing polypeptides both with mCitrine followed by repeats of S1 and S3 interchangeably repeated and separated with SAH linker and mCitrine followed by three repeats of S1 and three repeats of S3 separated with gs linker. d) TagBFP followed by three S2 and three S4 separated with SAH linker and TagBFP followed by three repeats of S2 and S4 interchangeably repeated and separated with gs linker. NIH-3T3 cells transfected with 100 ng of each plasmid. Scale bars, 10 μm .

Minor comments:

- Figure 1 is extremely complicated and challenging to follow. Please think of a way to simplify it, explain in more details in the text about each peptide and avoid abbreviations as much as possible.

We have extended the description of the schematic in the Figure description, to include what we mean by ‘binding network’ and that this leads to the phase separation: ‘Due to the use of two orthogonal weakly interacting CC pairs and their different arrangement both polypeptides cannot interact in a way to satisfy all the CC interaction domains, which means that one chain can interact with multiple other chains, leading to the formation of a polypeptide network, and phase separation.’

- For clarity, page 5 lines 133-135 should be moved to the paragraph on page 6 which focuses on the effect of linkers on LLPS.

We thank the reviewer for this suggestion, we have moved the said lines to page 6, as well as changed the Fig 2, to first describe liquid properties under sections c-f, and later the effect of linkers under g and h.

- Supplementary table 2: is the gray color code mentioned in the caption of the table supposed to be black? As no gray color appears in the sequences.

We thank the reviewer for catching this error, we have corrected the mistake in the revised version of the manuscript.

Reviewer #2 (Remarks to the Author):

Ramšak et al. present an elegant study on the design of liquid-liquid phase separating proteins using de novo coiled coil domains. Rather than focusing on engineering existing proteins, the authors highlight the power of bottom-up protein design for self-assembly. In this study, the authors use coiled coil dimers as modular building blocks for multivalent polypeptides that undergo LLPS. The authors empirically test different combinations of CC-forming domains while varying the number, lengths, and arrangements, as well as the rigidity and combinations of different linkers. In this way, they determine the appropriate valency and arrangement for phase separation. Perhaps uniquely in such engineered phase separating systems, the authors are able to test concepts such as mismatch between stickers in a biological setting, and rationally alter interaction affinities using coiled coil design rules. From a protein design perspective, this is perhaps the most interesting achievement of the study, as it demonstrates the capacity for the system to be understood in a way that most phase separating proteins cannot. Further, the authors use molecular dynamics to simulate the changes made to the designed molecules. Finally, the authors show that the designed molecules can be made orthogonal, and assembled and disassembled with previously well-characterised chemically inducible protein-protein interaction motifs (FKBP/FRB). Overall, the study shows the clear and growing role of protein design in LLPS, and we can only applaud their success in delivering de novo proteins forming highly dynamic liquid droplets in mammalian cells. The authors have developed some unique systems with their designs but have not used in vitro experiments at all to verify either their hypotheses around the coiled coil propensity or LLPS. We would encourage the authors to develop their study further, and make it a really strong contribution to the field, which could then be fully endorsed for publication in Nature Communications.

We thank the reviewer for the positive comment and suggestions to improve this manuscript. We address the reviewers' specific comments below:

1. The authors refer to established coiled coil dimer design rules for their dimer designs, however there is little detail on the selection of the specific sequences used (i.e., where did they come from, and why were these sequences chosen?). Can the authors clarify the design choices here?

We thank the reviewer for this comment. We have included a more thorough explanation and included it in the main text:

We designed an orthologous set of weakly interacting coiled-coil (CC) pairs, with adjustable affinities within to fit the range suitable for LLPS formation, sequences in Supplementary Table 1. The design was based on the same principles as before for the design of strongly interacting orthogonal pairs...

2. Further, the authors have made hypotheses regarding coiled coil propensity and orthogonality and its impact on LLPS, but these hypotheses have not been tested in vitro by standard biophysical methods. This is particularly poignant as the authors note the importance of weak interactions for LLPS, but they have not characterised the strength and specificity of the dimers they used. This would be very interesting for informing the relative strengths of

PPIs for phase separation, which is challenging to interrogate in other systems. Can the authors please provide biophysical characterisation of the utilised CC building blocks to support their rationalisation of interaction strength and valency in their designs.

The reviewer presents an important point, we have now analyzed the helical propensity of CC using CD and binding affinity of CC pairs with ITC.

Supplementary Figure 1: Circular dichroism (CD) spectra of a 1:1 mixture of CC.

Spectra comparison of single peptide and peptide pair for a, b) 3-heptad CC designed to have weak affinity; b, c) for 3-heptad CC designed with increased helical propensity and e, f) 4-heptad CC. g, h) Spectra depicting orthogonality, showcasing the comparison between peptide pairs and non-pair peptide combinations for g) 3-heptad peptides and h) 4-heptad peptides.

The CD analysis has provided evidence in support of our design hypothesis. Specifically, we found that the S1-S4 peptides exhibit minimal helical content when analyzed individually or in pairs. However, when we examined the pairs designed to have slightly higher affinity (the Sh

versions, with introduced Ala residues to increase their helical propensity), we observed a noticeable increase in their α -helical content. This increase is accentuated at lower temperatures. As per our design expectations, the 4-peptide coiled-coil (the Pf version) exhibited the highest helical content when appropriate pairs were combined. Once again, we observed a significant augmentation in helix formation when two coiled-coil pairs were present compared to when a single peptide was analyzed in isolation. Furthermore, we showed orthogonality between peptide pairs for Sh- and P- set, the designed pairs of peptides show more prominent helical content compared to mixtures of two peptides from distinct pairs.

These results not only validate our initial design hypothesis but also provide insights into the structural properties of these peptides and the blueprint for additional designs. We have included the above figure in the supplementary material – Supplementary Fig. 1.

Importantly, we have analyzed the weak binding affinity of peptide pairs using ITC:

Supplementary Figure 2: Binding analysis using isothermal titration calorimetry (ITC). For a) peptide pair S1h + S2h and b) S3h + S4h, which are 3-heptad CC with higher helical propensity. For peptide pair c) P5f + P6f and d) P13f + P14f, which are 4-heptad CC.

We have showed that the Pf set of peptide pairs have affinities in the range of a few μ M, while Sh versions have much lower affinities in the millimolar range. The S set of peptides have affinities in the range that could not be determined with ITC, most likely above 10 mM.

We have included the above figure in Supplementary material – Supplementary Fig. 2.

3. Why do the designer proteins require two orthogonal types of coiled coil dimers? The authors claim this is due to decreased branch propensity, but could it imply some crosstalk between supposedly orthogonal motifs? Is it pointing to the importance of promiscuous interactions of amphipathic helices rather than well-defined coiled coil dimers? Further, the reciprocal control for Fig. S1 (Cit(S1)6-SAH and BFP(S3)6-gs) is missing.

We appreciate the reviewer's insightful comment. Indeed, it is a matter of having the affinity in the appropriate range and nonmatching CCs do not seem to have sufficient affinity to lead to condensation. As described above we have determined the affinity between the interacting CCs that support the formation of LLPS. To additionally test this concern, we conducted an experiment in which we transfected cells with plasmids for two polypeptides, each containing the same CCs, either in the same arrangement (a, b) or in a differential arrangement (c, d), with each configuration featuring a different linker type. We did not observe any crosstalk or condensation in either scenario. These results strongly suggest that CC dimer interactions of the orthogonal motifs are indeed crucial for the formation of condensates in our system.

We acknowledge that this control was omitted from our initial submission, and we apologize for any oversight. We have now incorporated this information into our revised submission, and you can find the corresponding figure in the supplementary information - Supplementary Fig. 4.

Supplementary Figure 4: Polypeptides containing the same CC type do not form condensates in cells.

a) mCitrine fused to the repeats of S1 and S3 in an interchangeable arrangement and separated with SAH linker and mCitrine fused to the repeats of S1 and S3 in an interchangeable arrangement and separated with gs linker. b) TagBFP fused to three S2 and three S4 separated with SAH linker and TagBFP fused to three S2 and three S4 separated with gs linker. c) NIH-3T3 cells expressing polypeptides both with mCitrine followed by repeats of S1 and S3 interchangeably repeated and separated with SAH linker and mCitrine followed by three repeats of S1 and three repeats of S3 separated with gs linker. d) TagBFP followed by three S2 and three S4 separated with SAH linker and TagBFP followed by three repeats of S2 and S4 interchangeably repeated and separated with gs linker. NIH-3T3 cells transfected with 100 ng of each plasmid. Scale bars, 10 μm .

The following explanation regarding this figure was included in the main text: 'Furthermore, we showed that specific interactions between CC pairs are needed, as polypeptides containing the same CC type, but with different linker type and different arrangement did not form condensates (Supplementary Fig. 4).'

We have also included the reciprocal control for polypeptides with only a single CC type in Supplementary Fig. 1, that the reviewer noticed is missing.

We have cited it accordingly in the main text.

Additional text describing Supplementary Fig. 2 was included in the main text of the manuscript: 'Furthermore, we showed that specific interactions between CC pairs are needed, as polypeptides containing the same CC type, but with different linker type and different arrangement did not form condensates (Supplementary Fig. 2).'

4. S1-4h peptides are described as being designed by introduction of intrachain salt bridges. However, based on Table S1 they appear to differ from peptides S1-4 only by replacement of the b and c QS/SQ residues with AA. This does not appear to introduce additional intrachain salt bridges. Can the authors please confirm and clarify their design process here? Further, the authors seem to ignore a significant change in the protein hydrophobicity and H-bonding capacity caused by the QS->AA mutations that could be important for PPI and LLPS.

We thank the reviewer for catching this, it was a typographical mistake from our side and we apologize for the oversight on our part, since we have used both introduction of salt bridges and introduction of Ala residues to increase the helical propensity of CC-forming peptides.

We have corrected the mistake in the main text: 'Indeed, if CC segments with an increased helical propensity, achieved by the introduction of alanine residues, with higher helical propensity at positions *b* and *c*, substituting the glutamine and serine^{2,6}, were used (Fig. 3a, Supplementary table 1), we observed condensates even if both proteins had gs linkers...'

5. Fig. 2D seems to be contradicting the text. The constructs with SAH linkers do appear to form some kind of condensates. Could the authors review these data, and provide some rationalisation for these results?

Same issue as raised by Reviewer #1:

We apologize for any misunderstanding caused by the presentation of this experiment. We did not observe the formation of bona fide liquid condensates under those conditions. The images

presented in Fig. 2d gave the impression of condensed regions but this was rather due to the cellular structures. Furthermore, we want to clarify that in none of the cells, we observed the presence of round and droplet-like structures, as expected from liquid condensates. Therefore, we selected for presentation other images that were more representative and we show several additional images below.

To address these concerns and provide a more accurate representation of our findings, we have included pictures of more representative cells in our revised presentation (now Fig. 2h).

6. The molecular dynamics simulations used are interesting, and their relation to experiment has been performed well. However, we are not convinced that the simulations are acting predictively, which would be the ultimate goal of such a tool. Can the authors please provide some evidence that their simulations can behave predictively? For instance, the authors note that simulations of proteins with 12 coils suggested LLPS at a lower temperature. The experimental verification of this (or a similar in silico led hypothesis) would be extremely satisfying, and demonstrate that the MD simulations are able to act predictively.

The energy scale in the simulations does not exactly match experimental energies, but we believe they can offer quite compelling qualitative analysis and even predictions of some properties of LLPS. While 12 coil segment polypeptides did not form condensates even at lower temperature, we decided to test the simulation on a more complex system that can be constructed by the principles of CC-LLPS. For the revision we designed three-component systems while maintaining a blind approach, ensuring that our simulation team was unaware of the in vivo results. We employed one of the two-component condensate-forming systems, Cit(S1h-S3h)₃-gs and BFP(S2h)₃(S4h)₃-gs. We wanted to test if any of the additional polypeptide chains comprising the same modules could affect the formation of LLPS depending on the arrangement of the CC modules.

Notably, our observations the addition of a 4-coil polypeptide, Cit(S1h)₄-gs, which could bind to the chain featuring a 3+3 arrangement of CCs (BFP(S2h)₃(S4h)₃-gs), effectively prevented condensate formation (a). Conversely, the introduction of a 4-coil polypeptide, BFP(S2h)₄-gs, which interacts with the chain having four complementary S1h segments but in an alternating CC arrangement (Cit(S1h-S3h)₃-gs), did not prevent condensate formation (b). In both of those cases simulations aligned perfectly with the experimental results.

Furthermore, we explored an additional system featuring 4-heptad CCs. In this case, due to stronger interactions, Cit(P5f-P13f)₃-gs was able to undergo condensation with a partner possessing four coils of the same type, BFP(P6f)₄-gs. The alternating arrangement of Cit(P5f-P13f)₃-gs prevented dimer formation (d), which, in turn, led to condensation (c). In contrast, BFP(P6f)₃(P14f)₃ featured a segregated arrangement, allowing Cit(P5f)₄-gs to bind to three neighboring CC and thus preventing condensate formation (d). These results were also

consistent with simulations, further validating the agreement between in vivo and in silico results. Although there is a discrepancy in the LLPS properties of the 12-coil segment proteins at lower temperatures, we do not think this detracts from the predictive power demonstrated by simulation. We suspect this discrepancy is likely a result of the inexact energy scale between experiment and our coarse-grained simulation tool. Our ability to qualitatively match LLPS behavior between simulation and experiment is robust for 2-component and multi-component systems. Overall, our study demonstrates that our simulations align well with in vivo results, even in more intricate three-protein systems, underlying the value of those simulations.

The figure bellow is included in the supplementary information – Supplementary Fig. 15

Supplementary Figure 15: CC-LLPS condensate formation can be regulated by intentional formation of molecular dimers.

In all five subfigures (a-e), a schematic of the tested polypeptides along with their names are provided. Representative microscopy images shown as three image panels, with the mCit only channel (left-most), BFP only channel (middle), and the mCit-BFP merged channel (right-most). Scale bars are 10 μm . Results from predictive simulations are shown as density profile plots (left-most plot, with high density and low density regions indicating LLPS) and molecular cluster distributions (right-most plot), with high-aggregate numbers (40-100) indicating LLPS. In all cases, predictive simulations and experiments qualitatively agree on whether LLPS occurs. Solid lines represent the mean and shaded regions are the standard deviation from $N=3$ replicate simulations.

7. The use of FKBP/FRB as a chemically inducible heterodimeric PPI was perhaps prudent, but this system has already seen extensive use as a chemical inducer of, or recruiter to, protein condensates, and so it potentially does not bring a great deal of novelty to the study. As the rest of the study has covered the systematic design of coiled coils and their arrangement so comprehensively, this final section is somewhat underwhelming. The authors have made a very interesting orthogonal multi-component system with their coiled coil dimers, and it would potentially have been nice to see this applied to something instead to prove the utility of this system.

We appreciate the reviewer's thoughtful feedback and are grateful for their enthusiasm to see this system used in other applications. We acknowledge the use of FKBP/FRB may seem somewhat conventional, and we understand their concern about the novelty of this approach.

However, we want to emphasize that the inclusion of the FKBP/FRB system in our study serves as a valuable point of reference and comparison. By incorporating this widely used system into our work, we aim to demonstrate the versatility and utility of the coiled coil dimers in various contexts, including protein condensates. While it may appear somewhat underwhelming in isolation, it plays a crucial role in showcasing the broad applicability and orthogonality of our coiled coil-based multi-component system.

Furthermore, we are excited about the potential applications of our orthogonal multi-component system and agree with the reviewer's suggestion that it would be valuable to demonstrate its utility in a specific context. We are actively exploring such applications in our ongoing research and look forward to sharing those findings in future work.

Technical comments:

1. Reference 53 does not appear to direct to the appropriate literature (description of a coarse-grained framework, line 203 of the manuscript PDF). Could the authors please confirm that the correct literature is referenced here?

We thank the reviewer for catching this error, we have corrected it in the revised version of the manuscript.

2. Overall, while the design framework used is overall comprehensive, it is occasionally challenging to follow in the manuscript and could benefit from restructuring for clarity. Mentioning the concept of “stickers and spacers” (which these designs clearly resemble) would also be beneficial as it has been widely used to rationalize LLPS of proteins. A summary table of designs in the supplementary information could help direct the reader.

To enhance the reader's experience and facilitate a better understanding of our manuscript, we have included a table that provides a comprehensive overview of all the various polypeptides used in our study. Additionally, we have cross-referenced each polypeptide with the corresponding figure number where it is presented. We acknowledge that the numerous similar but distinct constructs in our study can sometimes be challenging to differentiate, and we believe that this table will serve as a valuable reference to help readers navigate the content more easily.

We also included a mention of the stickers and spacers framework in the introduction part: 'A stickers and spacers framework has been developed, where stickers represent the interacting domains and spacers the regions separating them'^{7,8}

We note that that the CC interaction domains function as stickers. 'To provide multivalent interactions driving phase separation, each polypeptide chain must contain several interacting CC segments acting as stickers'^{7,8} that are concatenated into a single chain.'

REFERENCES

1. Gradišar, H. & Jerala, R. De novo design of orthogonal peptide pairs forming parallel coiled-coil heterodimers. *Journal of Peptide Science* **17**, 100–106 (2011).
2. Drobnak, I., Gradišar, H., Ljubetič, A., Merljak, E. & Jerala, R. Modulation of Coiled-Coil Dimer Stability through Surface Residues while Preserving Pairing Specificity. *J Am Chem Soc* **139**, (2017).
3. Plaper, T. *et al.* Designed allosteric protein logic. doi:10.1101/2022.06.03.494683.
4. Gradišar, H. *et al.* Design of a single-chain polypeptide tetrahedron assembled from coiled-coil segments. *Nat Chem Biol* **9**, 362–366 (2013).
5. Banani, S. F. *et al.* Compositional Control of Phase-Separated Cellular Bodies. *Cell* **166**, 651 (2016).
6. Pace, C. N. & Scholtz, J. M. A helix propensity scale based on experimental studies of peptides and proteins. *Biophysical Journal* vol. 75 422–427 Preprint at [https://doi.org/10.1016/s0006-3495\(98\)77529-0](https://doi.org/10.1016/s0006-3495(98)77529-0) (1998).
7. Choi, J.-M., Holehouse, A. S. & Pappu, R. V. Physical Principles Underlying the Complex Biology of Intracellular Phase Transitions. (2020) doi:10.1146/annurev-biophys-121219.
8. Choi, J. M., Dar, F. & Pappu, R. V. LASSI: A lattice model for simulating phase transitions of multivalent proteins. *PLoS Comput Biol* **15**, (2019).

Reviewers' Comments:

Reviewer #1:

Remarks to the Author:

The authors have fully addressed my comments and added convincing new data, which support their conclusions. The CD and ITC results are insightful and valuable for the paper. I now recommend on the publication of the work in Nat. Commun.

Reviewer #2:

Remarks to the Author:

We thank the authors for their response to our questions. We'd like to particularly commend them on their comprehensive response, which has added a great deal of interesting data and concepts to the study. Overall, they have produced an excellent piece of work. We have only a few minor queries regarding the new data that they have presented, after which we fully endorse its publication in Nature Communications.

1. The CD data for the peptides supports the design approach undertaken in the manuscript.

However, the presented data needs to be corrected before it can be accepted for publication:

a. The conditions used (such as peptide concentrations), though present in the Methods, should also be mentioned in the captions to the figures.

b. The data seem to have signal normalisation problem. For example, MRE values at 222nm on most panels are too high for the observed MRE208/MRE222 ratio that suggests errors in peptide concentration. Also, some of MRE222 values seem to be greater than theoretically predicted for such short peptides which points to the same normalisation problem.

c. The labels to the Y-axes (MRE) should be fixed.

2. The ITC data presents a good piece of evidence that the affinities of the peptides follow the expected trend introduced by design. However, for the weakly interacting pairs of peptides (S1h-S2h, S3h-S4h), due to the low affinities neither of the two plateaus were reached which suggests large fitting errors that should be presented along with the values.

3. The question #3 from the first revision (Why do the designer proteins require two orthogonal types of coiled coil dimers?) wasn't fully addressed and requires additional clarification. The crosstalk is related to the interaction between orthogonal CC dimerization motifs rather than the two proteins. The CD data demonstrates orthogonality of different pairs of peptides but the role of having two types of interacting stickers in the formation of condensates remains unclear. Having orthogonal motifs should decrease the multivalency because each motif can only interact with half as many partner motifs as it has in the control systems with only one pair of motifs.

We appreciate that this observation is puzzling and may not have a clear answer at this stage, but it might be a universal design rule applicable to all sticker-spacer-sticker-etc systems regardless of the specific nature of the sticker motifs.

Reviewer #3:

None

REVIEWERS' COMMENTS

Reviewer #1 (Remarks to the Author):

The authors have fully addressed my comments and added convincing new data, which support their conclusions. The CD and ITC results are insightful and valuable for the paper. I now recommend on the publication of the work in Nat. Commun.

Reviewer #2 (Remarks to the Author):

We thank the authors for their response to our questions. We'd like to particularly commend them on their comprehensive response, which has added a great deal of interesting data and concepts to the study. Overall, they have produced an excellent piece of work. We have only a few minor queries regarding the new data that they have presented, after which we fully endorse its publication in Nature Communications.

1. The CD data for the peptides supports the design approach undertaken in the manuscript. However, the presented data needs to be corrected before it can be accepted for publication:

a. The conditions used (such as peptide concentrations), though present in the Methods, should also be mentioned in the captions to the figures.

b. The data seem to have signal normalisation problem. For example, MRE values at 222nm on most panels are too high for the observed MRE208/MRE222 ratio that suggests errors in peptide concentration. Also, some of MRE222 values seem to be greater than theoretically predicted for such short peptides which points to the same normalisation problem.

c. The labels to the Y-axes (MRE) should be fixed.

2. The ITC data presents a good piece of evidence that the affinities of the peptides follow the expected trend introduced by design. However, for the weakly interacting pairs of peptides (S1h-S2h, S3h-S4h), due to the low affinities neither of the two plateaus were reached which suggests large fitting errors that should be presented along with the values.

We thank the reviewers for their comments. We have corrected the figures on CD, added additional information to Figure captions and provided an estimate of uncertainty for the ITC curve.

The question #3 from the first revision (Why do the designer proteins require two orthogonal types of coiled coil dimers?) wasn't fully addressed and requires additional clarification. The crosstalk is related to the interaction between orthogonal CC dimerization motifs rather than the two proteins. The CD data demonstrates orthogonality of different pairs of peptides but the role of having two types of interacting stickers in the formation of condensates remains unclear. Having orthogonal motifs should decrease the multivalency because each motif can only interact with half as many partner motifs as it has in the control systems with only one pair of motifs.

We appreciate that this observation is puzzling and may not have a clear answer at this stage, but it might be a universal design rule applicable to all sticker-spacer-sticker-etc systems regardless of the specific nature of the sticker motifs.

In the initial reply to the question #3 we replied on the issue of orthogonality but indeed did not respond adequately on the question why two different orthogonal types of coiled coil dimers are required for which we offer the following explanation.

We reasoned that in case of using a single pair of CC peptides, where each of the two chains comprising only a single type of a CC motif (for example AAAA and aaaa for an Aa type CC heterodimer), this could end with the formation of dimers (if the number of CC modules is the same in both chains) or formation of staggered linear polymers, where each chain would typically interact with maximally two complementary chains. This arrangement would not be conducive for the formation of a highly crosslinked phase, which likely requires percolation of molecular connections, which can be achieved when each chain interacts with multiple chains of the complementary type to form a network of (transient) interactions. A chain with a single interacting CC sticker can only form a linear pattern (e.g. AAAA), except perhaps for the introduction of inert linkers of different lengths that could result in a mismatch between the two chains.

The introduction of two (or more) pairs of CC stickers, on the other hand, enables the introduction of different arrangements of CC segments, e.g. in a pattern AAABBB (clustered) and ababab (alternating) for Aa and Bb types of dimers. In this case simple heterodimers or staggered linear

chains cannot have fully satisfied binding sites but facilitates interactions with additional chains, leading to the formation of highly networked clusters.

This principle is also valid for phase separation composed of a single chain presented in a manuscript, where interaction between two types of homodimeric CC stickers occurs in different orientations which facilitates high networking instead of linear connections. In this case, we demonstrated that only an asymmetric pattern APAPPAPAPP is able to phase separate but not a clustered arrangement AAAAAPPPPP, comprising the same types and numbers of CC segments but in a different arrangement.

We therefore believe that multivalency by itself is not sufficient to phase separate polypeptides composed of CC-dimer forming segments but the arrangement should also hinder interactions between multiple stickers on pairs of chains, as this increases the interaction affinity between the two chains and hinders rapid dissociation required for the liquid phase.

We hope this explains our rationale for the design and explanation of the results.